# Increased urinary creatinine during hibernation and day roosting in the Eastern bent-winged bat (*Miniopterus fuliginosus*) in Korea

Heungjin Ryu [1,2], Kodzue Kinoshita[3], Sungbae Joo[2], Yu-Seong Choi[4] & Sun-Sook Kim [2✉]

Torpor and arousal cycles, both daily and seasonal (e.g. hibernation), are crucial for small mammals, including bats, to maintain the energy and water balance. The alternation between torpor and arousal leads to metabolic changes, leaving traceable evidence of metabolic wastes in urine. In this study we investigated urinary creatinine and acetoacetate (a ketone body) in the Eastern bent-wing bat (*Miniopterus fuliginosus*) in Mungyeong, South Korea. We found an increase in urinary creatinine during torpor in summer, indicating changes in renal water reabsorption rates during the active season. Although we could not confirm ketonuria in hibernating bats due to a methodological limitation caused by the small amount of urine, we verified an increase in urinary creatinine concentration during hibernation. This finding suggests that managing water stress resulting from evaporative water loss is one of key reasons for arousal during hibernation in Eastern bent-wing bats.

[1] Department of Social Informatics, Kyoto University, Yoshidahonmachi, Sakyo-ku, Kyoto 606-8501, Japan. [2] National Institute of Ecology, Geumgang-ro 1210, Maseo-myeon, Seocheon, Chungnam 33657, Republic of Korea. [3] Graduate School of Asian and African Area Studies, Kyoto University, Yoshidahonmachi, Sakyo-ku, Kyoto 606-8501, Japan. [4] National Migratory Birds Center, National Institute of Biological Resources, Incheon 22689, Republic of Korea. ✉email: sskim108@gmail.com

Physiological ecology investigates physiological changes and their underlying mechanisms in organisms under natural conditions[1,2]. Physiological processes, in response to an animal's surrounding environment, play a crucial role in maintaining the animal's physical condition along with behavior. Therefore, understanding the mechanisms underlying these processes can help us comprehend how animals have evolved to cope with environmental constraints. Small mammals are important subjects in physiological ecology due to their diverse behavioral and physiological adaptations for survival and reproduction, which address the limited fat storage capacity resulting from their small body size[3–5]. Hibernation is a vital adaptation for the survival of small mammals during winter when food supplies are limited. Bats, in particular, exhibit various behavioral strategies and physiological adaptations to cope with challenges such as evaporative water loss during hibernation, due to their small body size and greater surface-to-volume ratio (SVR) compared to other small mammals[6–9].

Along with maintaining water and energy balance, hibernating small mammals need to remove accumulated metabolic wastes. Fat is an important energy source for hibernating small mammals[10–12]. However, intensive fat metabolism leads to an increase in serum accumulation of ketone bodies[12]. As an increase in serum ketone bodies influences serum pH, it needs to be properly controlled. However, due to the decrease of blood flow to the kidneys during torpor, the glomerular filtration rate also decreases significantly[5,13]. Short arousal intervals between torpor bouts provide crucial opportunities for hibernating small mammals to remove serum accumulation of metabolic wastes such as ketone bodies via urine[5]. As glomerular filtration increases during arousal, it is expected that the urinary concentration of ketone bodies and other metabolic wastes will also increase.

In addition to removing metabolic wastes, maintaining water balance during hibernation is essential for successful hibernation. To overcome the shortage of water supply, hibernating small mammals exhibit various behavioral strategies and physiological adaptations. For example, thirteen-lined ground squirrels (*Ictidomys tridecemlineatus*) can maintain low extracellular fluid osmolality during hibernation, which suppresses thirst that can drive a water-seeking response[14]. European ground squirrels (*Spermophilus citellus*) may suppress urine production during torpor, helping them reduce water loss and arousal frequency during hibernation[15]. In bats, metabolic water formation is insufficient to compensate for evaporative water loss, leading them to occasionally arouse to drink water during hibernation[6,16,17].

Bats are unique among small mammals in that they exhibit various adaptations for flight, including reduced body mass and feeding on energy-rich diets[18]. Bat wings are the most important adaptation for their aerial lifestyle. However, their wings can increase the SVR compared to that of non-volant mammals of similar body size. Although the presence of lipids in bat wings may prevent significant evaporative water loss[19], the large SVR of bats can also increase evaporative water loss from the skin[20]. The difficulty in maintaining water balance is most prominent during hibernation when water and food intake are limited. To reduce the cost of evaporative water loss, many small insectivorous bats in the temperate zone select a hibernaculum with low ambient temperature (TA) and high relative humidity (RH)[21,22]. However, relying on such abiotic factors may not be sufficient for maintaining water balance during hibernation, necessitating behavioral and/or physiological solutions[6,8,9].

To mitigate the disadvantages of small body size during winter, hibernating bats adopt several behavioral strategies, including huddling, winter feeding, and changing hibernation sites[6,21,23,24]. However, tracking individual behaviors is challenging due to the difficulty of direct observation. Even with remote devices, like infrared cameras, behavioral observation can cause disturbance to hibernating bats, as those devices need maintenance by human operators. An alternative to direct observation is investigating physiological markers from their excretion to estimate behaviors and physiological processes. Given that other hibernating small mammals resume renal filtration during arousal[5,25], hibernating bats may exhibit similar behavior. If this is the case, any arousal during hibernation will leave traceable evidence in their excretions, including urine. However, data on physiological parameters in urine that may reflect the behavioral response of hibernating bats are limited and have yielded mixed results. For example, little brown bats (*Myotis velifer*) do not produce hypotonic urine during hibernation[26,27], whereas greater horseshoe bats do[28]. Such inter-species differences may reflect significant species-specific social and environmental differences during hibernation. However, the available data is limited to provide a comprehensive understanding of the behavioral and physiological adaptations of hibernating bats.

Urine analysis is widely used to non-invasively investigate the physiological status and metabolism of mammals[12,29,30]. Compared with fecal samples, urine samples offer advantages in terms of ease of use and quantitative measurement[30]. They require a smaller amount for analysis and involve fewer handling processes. They can also be stored at room temperature for several months if properly dried[31–33]. This is a significant advantage for wildlife studies that require long-term stays in remote field areas where access to electricity and equipment for sample storing is often limited. In addition, the concentration of hormones or other metabolites in urine can be adjusted relative to urine concentrations or other references[34]. This adjustment is important given the various social and environmental variables that can influence the degree of dehydration in wild animals. Although the technique has yet to be widely applied, some pioneering studies have demonstrated the usefulness of urinary analysis in bats. For example, it has been shown that urine concentration reflects the hydration status and energy expenditure in both insectivorous and frugivorous bats[35–38]. Nevertheless, there have been few examples of urinary analysis of insectivorous bats, likely due to the small amount of urine they produce[27,28].

Creatinine is a metabolic product of creatine in muscle and is constantly excreted in urine[39]. It has long been used to adjust substrate concentration in urine as it correlates well with urine concentration[40]. Using this characteristic, urinary substance concentrations are adjusted by creatinine concentration to avoid the effect of urine concentration on urinary substance concentrations[34]. Acetoacetate is one of three ketone bodies found in urine when intensive stored fat metabolism occurs due to fasting or limited food intake[41].

In this study, we analyzed creatinine and acetoacetate in urine samples from a population of Eastern bent-winged bat (*Miniopterus fuliginosus*) in Mungyeong, South Korea. Eastern bent-winged bats are small insectivorous bats distributed in Korea, Japan, and China[42,43]. They form large hibernation clusters consisting of several hundred to thousands of individuals. Pregnant females migrate for breeding and form a substantial breeding colony[42]. They serve as excellent subjects for studying physiological ecology in the wild. In particular, the population in this study forms a large colony of several hundred bats year-round, ensuring a large sample size, which is often challenging in field studies. We can also capture them by hand for sampling during winter without using any special equipment as they form a large cluster at a hand-reaching distance from the ground. Although the amount of urine collected is small (around 20 to 30 μl), they also urinate easily during handling, which helps reduce the time required for urine sample collection.

We hypothesized that Eastern bent-winged bats increase fat metabolism, resulting in the accumulation of serum ketone bodies during hibernation. A higher concentration of ketone bodies (ketonuria), particularly acetoacetate in urine, is expected during hibernation compared with the non-hibernation season. We also hypothesized that hibernating Eastern bent-winged bats experience water-stress during hibernation. We expected that hibernating bats would increase glomerular water reabsorption to compensate for evaporative water loss. An increase in glomerular water reabsorption would result in a higher urinary creatinine concentration during winter (hibernation season) compared with spring, summer, and autumn (active season). To test these hypotheses, we collected urine samples from Eastern bent-winged bats once a month from June to March. We then measured the creatinine and acetoacetate concentration of urine samples and investigated factors that may influence concentration variations including body mass and forearm length (as a proxy for surface area), as well as season and sex. Based on our data, we discussed the potential causes of the seasonal variations in urinary metabolites (creatinine and acetoacetate) and their implications for the physiological adaptations of bats.

## Results

We captured 530 Eastern bent-winged bats (165 females and 365 males) at a natural cave (Sukkul, A in Supplementary Fig. 1), and an abandoned mineshaft (Hogye, B in Supplementary Fig. 1) in Mungyeong, South Korea from July 2017 to March 2018. We collected 162 urine samples from 161 bats (Sukkul: 131, Hogye: 30, Table 1). Only one bat was sampled twice in October and December. In November, we could only capture five bats at dusk and collected urine from three bats, although we also conducted a mist net survey at dawn. The five bats were caught outside of the mist net at dusk implying that they were coming into the cave from outside, not going out from the cave. This was different from what we observed from July to October when most bats were caught inside of the mist net at dusk while they were exiting the cave. We did not observe any bats around our mist net at dawn in November as well. Therefore, we considered the active season to be from June to October, the hibernation season from December to March, and November a transient period.

We measured the body mass and forearm length of 119 bats among 133 urine-collected bats from July to October (raw data provided in the Supplementary Data), as 14 bats escaped during our body size measurements. We could measure the body size of two bats among three urine-collected bats in November. We did not measure the body sizes of hibernating bats from December to

March to minimize our time in the hibernaculum. We also recorded ambient temperature (TA) and relative humidity (RH) at three locations in Sukkul cave every hour from June 1, 2017, to April 24, 2018.

The average daily TA varied depending on the location within Sukkul cave (Supplementary Fig. 2). The entrance showed the largest variations in temperature (10.34 ± 10.42 °C; mean ± SD) in the range of −10.5 to 27.9 °C (Supplementary Fig. 2a, b), reflecting the TA at the nearest weather station (8.2 km from Sukkul). In the middle of the cave, the variations were smaller (11.25 ± 7.04 °C, range of −2.2 to 23.1 °C; Supplementary Fig. 2c). At the end of the cave where bats were hibernating, the TA was stable throughout the study period (12.09 ± 2.29 °C, range of 6.6 to 15.1 °C). The average daily RH also varied between the locations within the cave (entrance: 72.28 ± 17.24%, range of 33.4–97.9%, middle: 64.86 ± 14.9%, range of 38.0–87.2%, end: 76.77 ± 24.16%, range of 35.5–98.5%). The largest SD at the end of the cave was due to the sudden decrease in RH in August and the increase in November. The sudden changes in RH, recorded at the end of the cave (as shown in Supplementary Fig. 2d), raised questions about potential logger malfunctions. To prevent data misinterpretation, we decided to exclude these data from further analysis. Nevertheless, it was evident that the RH variation decreased from the cave's entrance to its end, similar to the pattern observed in the TA.

Bats captured at dawn (51 in total) upon their return to the roost were heavier than those captured at dusk (68 in total) during the active season, with masses of 16.03 ± 1.42 g and 14.81 ± 1.34 g respectively. A Welch's t-test ($t = 4.78$, $df = 104.56$, $p < 0.001$) demonstrated this difference is statistically significant. However, bats captured at dawn were not always heavier than those captured at dusk, as observed in October (refer to Supplementary Table 1; Fig. 1a). While there was a decreasing trend in the body mass of bats captured at dusk until September, this trend reversed in October. The forearm length, which serves as a proxy for surface area, showed no significant difference between bats captured at different times (Welch's t-test; $t = -0.29$, $df = 99.14$, $p = 0.77$), indicating that our data did not exhibit a significant sampling bias for wing size between different capture time (Supplementary Table 1).

**Urinary creatinine**. Creatinine was successfully detected in the 162 diluted urine samples. The urinary creatinine analysis yielded a very low intra-assay coefficient of variation (CV; 2.40 ± 2.13%), demonstrating the reliability of our urinary creatinine assay for bat urine samples. Monthly variations in creatinine concentration showed a negative trend with changes in body mass from July to October, particularly in bats captured at dusk (Fig. 1b). The average creatinine concentration from the urine during the hibernation season (26 samples from Dec to Mar) was almost double that in the active season (133 samples from Jun to Oct): 0.50 ± 0.31 mg/ml versus 0.29 ± 0.23 mg/ml, respectively.

We conducted a multiple linear regression (Model 1) to test the effect of the season, sex, and sampling location on the urinary creatinine (adjusted $R^2 = 0.101$, $F (3, 155) = 6.898$, $p < 0.001$). As a result of Model 1, we found higher creatinine concentrations from samples collected in the hibernation season than those collected in the active season (Table 2 and Fig. 2). There was no apparent sex difference or location difference in creatinine concentration (Fig. 2 and Table 2).

We further investigated the effect of body mass, forearm length, captured time, and sex on creatinine concentration during the active season using a generalized linear mixed model (Model 2). We included an interaction term between body mass and forearm length to test for potential dependency between the two

**Table 1 The number of urine samples and the date of urine collection.**

| Months | Creatinine | Acetoacetate | Date of urine collection |
|---|---|---|---|
| 2017. 07 | 26 | 21 | 7/13–14 |
| 2017. 08 | 34 | 20 | 8/10–12 |
| 2017. 09 | 39 | 33 | 9/19–20 |
| 2017. 10 | 34 | 30 | 10/18–19 |
| 2017. 11 | 3 | 2 | 11/15 |
| 2017. 12 | 7 | 7 | 12/20 |
| 2018. 01 | 14 | 11 | 1/22 |
| 2018. 02 | NA | NA | NA |
| 2018. 03 | 5 | 3 | 3/6 |
| Sum | 162 | 127 | |

In total, 162 samples were collected. Due to the small amount of collected urine, acetoacetate concentration could only be measured for 127 samples. In February 2018, we did not conduct sampling to minimize disturbance to hibernating bats. NA not available.

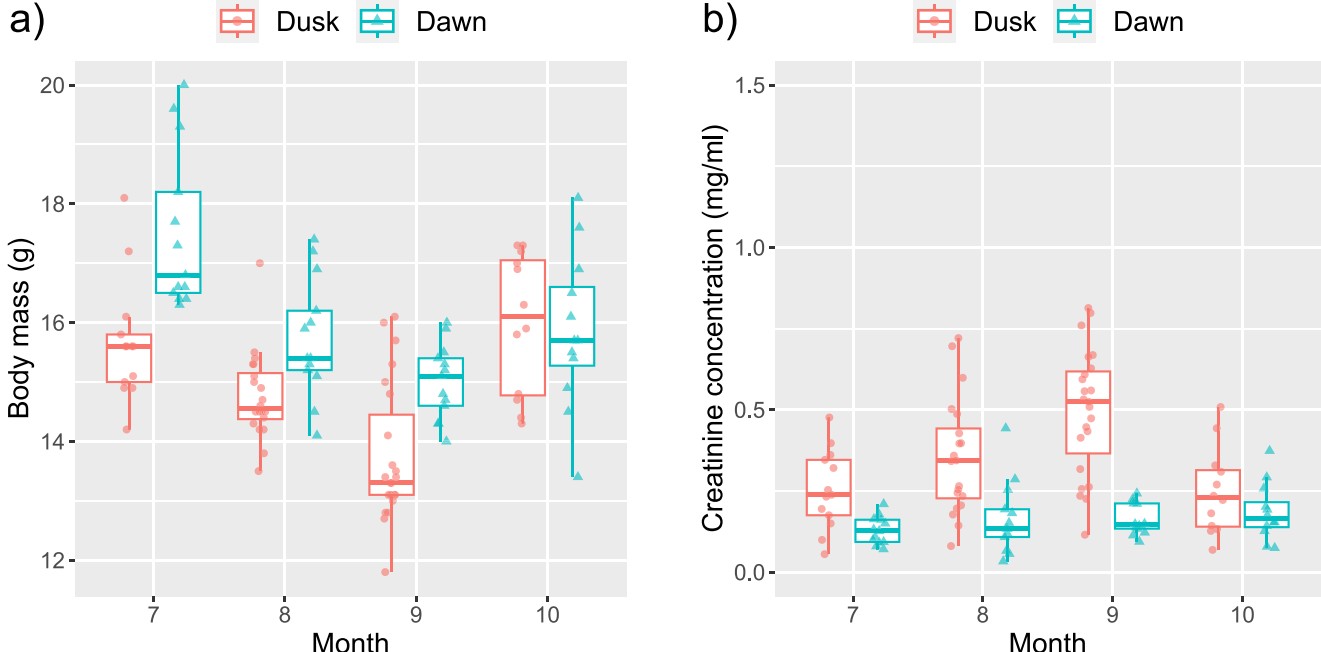

**Fig. 1 Monthly changes in body mass and urinary creatinine concentration in relation to captured time (feeding status). a** The body mass of the bats decreased until September and then increased in October. **b** The urinary creatinine concentration from the bats emerged from their daily roost at dusk (unfed) increased until September but decreased in October. However, there was no such trend observed in the urine collected from the bats that came back to their roost at dawn (fed). The upper and lower edges of each box represent the 75th and 25th percentiles, respectively. The line within the box indicates the median, the whiskers show 1.5 times the interquartile range, and the symbols represent individual data points ($N = 119$).

**Table 2 Linear model (Model 1) and linear mixed model (Model 2) summary results.**

|         | Predictor variables       | Esti. | SE   | df     | t     | p          | 95% CI        |
|---------|---------------------------|-------|------|--------|-------|------------|---------------|
| Model 1 | Intercept                 | −1.41 | 0.16 | -      | −8.92 | <0.001***  | −1.72, −1.09  |
|         | Sex (male)                | −0.07 | 0.12 | -      | −0.60 | 0.550      | −0.32, 0.17   |
|         | Season (hibernation)      | 0.67  | 0.15 | -      | 4.34  | <0.001***  | 0.36, 0.97    |
|         | Cave (Sukkul)             | 0.00  | 0.14 | -      | −0.00 | 0.998      | −0.29, 0.29   |
| Model 2 | Intercept                 | −1.20 | 0.14 | 4.16   | −8.55 | <0.001***  | −1.47, −0.93  |
|         | Body mass                 | −0.20 | 0.06 | 112.54 | −3.56 | <0.001***  | −0.31, −0.09  |
|         | Forearm length            | 0.11  | 0.05 | 112.67 | 2.17  | 0.032*     | 0.01, 0.20    |
|         | Sex (male)                | −0.07 | 0.12 | 112.61 | −0.57 | 0.567      | −0.29, 0.16   |
|         | Captured time (dawn)      | −0.71 | 0.12 | 82.56  | −5.99 | <0.001***  | −0.93, −0.47  |
|         | BW: FAL                   | 0.13  | 0.05 | 112.01 | 2.55  | 0.012*     | 0.03, 0.22    |

In Model 1, creatinine concentrations were compared between sexes (male vs. female), seasons (hibernation vs. active), and caves (Sukkul vs. Hogye). In Model 2, the effects of sex, season, captured time (dawn vs. dusk), body mass, forearm length, and interaction between body mass and forearm length on urinary creatinine concentrations were examined. Body mass and forearm length were standardized ((x-mean)/SD) and creatinine concentrations were naturally log-transformed. Esti.: estimates, ***$p < 0.001$, *$p < 0.05$, BW: FAL is the interaction term between body mass and forearm length. () in predictors represent the reference factor.

variables—body mass as a proxy for body volume and forearm length as a proxy for body surface area. In the model, the body mass and forearm length were standardized (see methods). We found that individuals with heavier body masses had lower urinary creatinine concentrations during the active season (Fig. 3a and Table 2). Although Model 2 showed that urinary creatinine concentration increased with forearm length (Table 2), as shown in Fig. 3b, the slope and broadly scattered data points indicate that the effect of forearm length was not definitive. We also found that bats emerging from their daily roost at dusk for feeding (presumably unfed bats), had higher creatinine concentrations (Fig. 3c). However, there was no sex difference in urinary creatinine concentrations (Fig. 3d).

In addition, despite the aforementioned significant main effects in Model 2, the significant interaction term suggests interdependency between the effect of body mass and forearm length on urinary creatinine. To clarify this interdependence and its

potential effect on urinary creatinine, we conducted a post-hock investigation of the effect of body mass on urinary creatinine concentration relative to forearm length in Model 2. We visualized the interaction term of Model 2 by fitting the effect of the body mass on urinary creatinine with three different forearm lengths (mean forearm length and mean ± 1 SD) – simple slopes analysis[44]. As shown in Fig. 4, we found that heavier bats produced lower creatinine-concentrated urine. However, this negative effect of body mass on creatinine concentration was stronger for the bats with shorter forearms—heavier bats with shorter forearms had significantly lower urinary creatinine concentration (Fig. 4).

**Urinary acetoacetate (a ketone body)**. Acetoacetate concentrations from 127 urine samples were notably low, resulting in a high intra-assay CV. Among these, 62 samples resulted in intra-assay

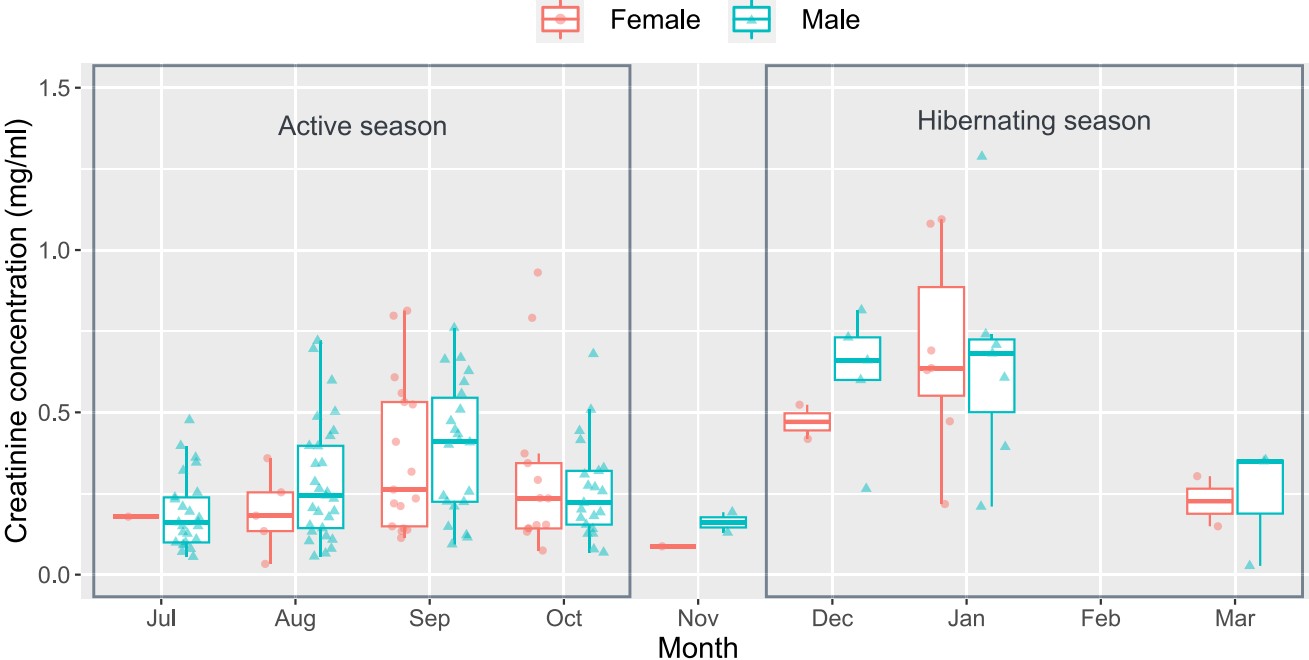

**Fig. 2 Comparison of urinary creatinine concentration between seasons and sexes.** The urinary creatinine concentration was found to be higher during hibernation than during the active season. There was no observable difference in creatinine concentration between sexes over months. The upper and lower edges of each box represent the 75th and 25th percentiles, respectively. The line within the box indicates the median, the whiskers show 1.5 times the interquartile range, and the symbols represent individual data points ($N = 162$).

CVs ranging from 0% to 50% (19.45 ± 12.83%; Supplementary Data), while the rest were outside these ranges. None of the samples showed acetoacetate concentrations higher than the detection limit of the acetoacetate assay kit—5.0 nmol per well (Fig. 5a). The median acetoacetate concentration from the 20 filtered and 20 unfiltered urine samples was 0.245 and 0.168 µmol/ml, respectively (Fig. 5b). Filtering did not elevate the sample concentration above the detection limit. Consequently, we did not proceed with further analysis of urinary acetoacetate concentrations.

**Discussion**

In this study, we examined two urinary metabolic markers, creatinine, and acetoacetate, to understand the physiological changes in Eastern bent-winged bats during hibernation and the active season. We were unable to find any valid acetoacetate (ketone bodies) concentrations in the urine, likely due to our high dilution rates. However, we did find seasonal variations in urinary creatinine, indicating more concentrated urine production during hibernation. This finding suggests that Eastern bent-winged bats produce concentrated urine due to limited water intake associated with prolonged torpor bouts during hibernation. This is in line with findings for other insectivorous bats in the temperate zone, where arousal of hibernating individuals is linked to evaporative water loss[6,16,20]. Nevertheless, there was a large difference in the number of urine samples between the hibernation season and the active season in this study. This discrepancy in sample size between the two seasons could introduce a sampling bias and should be taken into account before drawing comprehensive conclusions on this topic.

The accumulation of ketone bodies from intensive fat metabolism during hibernation can decrease serum pH. To balance the serum pH level, hibernating bats may arouse to transfer ketone bodies from serum to urine[45]. Although the current study aimed to clarify this possibility, we could not confirm it as our urinary

acetoacetate analysis did not yield meaningful results. There could be two possible explanations for our result. One is that the high dilution rate (at least 20 times) due to the limited amount of urine samples made it impossible to measure the urinary acetoacetate concentration. As the detection limit (sensitivity) of the kit was 5 nmol per well, a dilution rate of at least 20x in this study made it impossible to detect acetoacetate even if there was enough acetoacetate in undiluted urine samples. The other possibility is that the Eastern bent-winged bats in this study did not exhibit ketonuria during hibernation and active season. This possibility can be tested using other methods including paper strip tests with undiluted urine samples or mixing up several urine samples collected in the same season to make enough volume for urine analysis.

We demonstrated that urinary creatinine concentration in Eastern bent-winged bats changes over time, particularly during day roosting in the active season (July to October). These changes in urinary creatinine during the active season are not likely a result of changes in muscle mass, as their body mass decreased while urinary creatinine increased through September (Fig. 1a, b). We lack data to further investigate the behavioral causation influencing urinary creatinine from the bats that emerged from the cave after day roosting. However, it is possible that changes in torpor duration during day roosting in the active season may influence such creatinine variations between months[7,11]. When bats increase arousal time, the renal filtration rate will increase as their body temperature rises. This increase in renal filtration during the daytime will result in water filtration or reabsorption in the nephron, depending on individual body water balance. For example, bats with lower body water will increase water reabsorption, which will then increase the accumulation of creatinine in urine. Conversely, bats with sufficient body water will produce more urine by water filtration, which increases urine volume and results in lower urinary creatinine concentration. As we performed spot urine collection, this may not represent the entire urine production during day roosting. Additionally, we could not

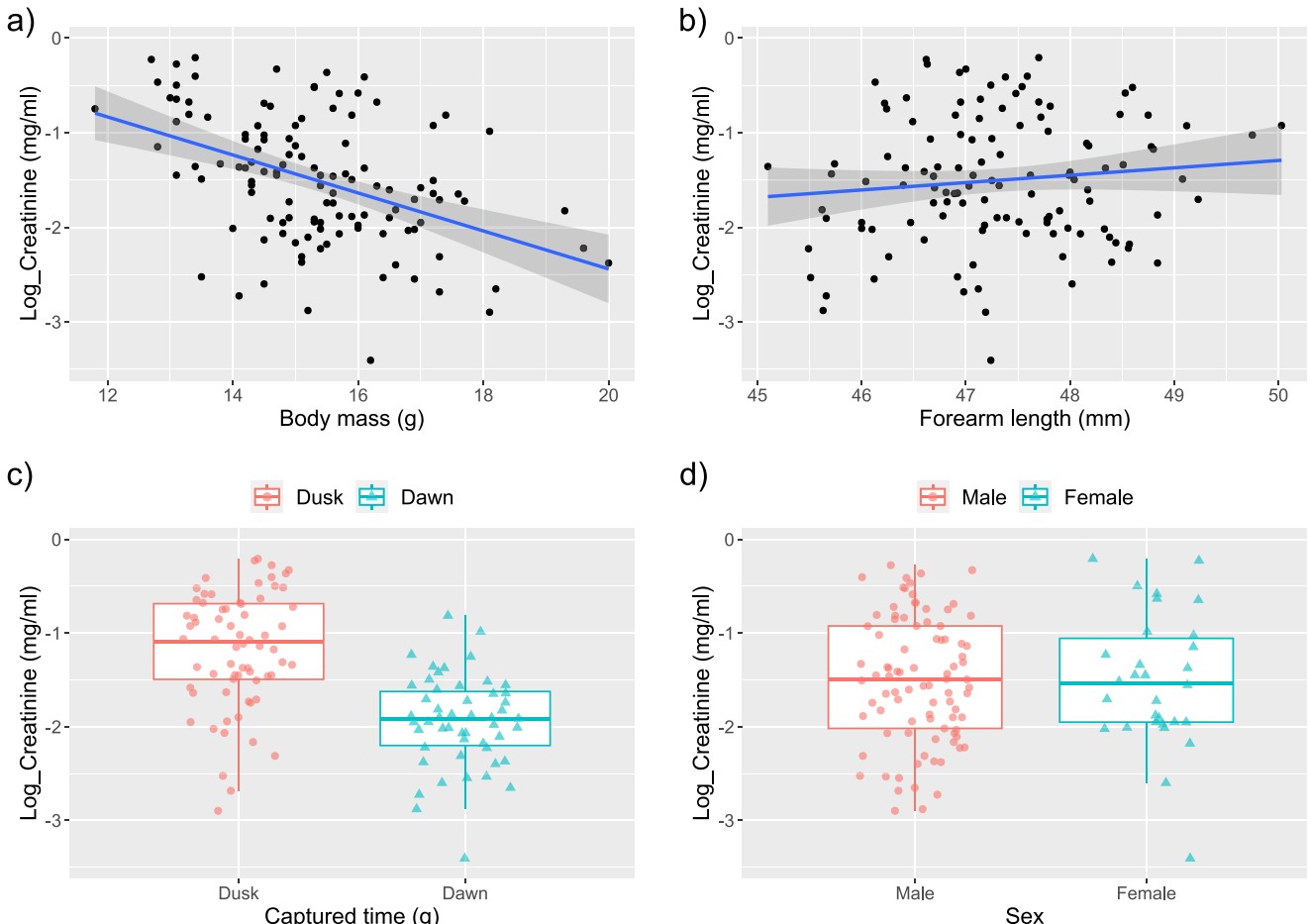

**Fig. 3 Effect of body mass, forearm length, sex, and captured time (feeding status) on urinary creatinine concentration during the active season.** The effect of body mass, forearm length, and captured time on the urinary creatinine (natural log-transformed) were significant. However, Model 2 (Table 2) also revealed a significant interaction between body mass and forearm length. **a** There was a negative correlation between body mass and creatinine concentration. **b** Despite the positive correlation between forearm length and urinary creatinine concentration, the data points were broadly scattered along the regression line. **c** Bats captured at dusk demonstrated higher urinary creatinine levels. **d** No difference in urinary creatinine concentration was found between the sexes. The lines in each figure represent regression lines from simple regressions to aid visual interpretation, and the gray bands around these lines denote 95% confidence intervals. The upper and lower edges of each box represent the 75th and 25th percentiles, respectively. The line within the box indicates the median, and the whiskers show 1.5 times the interquartile range. The symbols in each figure represent individual data points ($N = 119$).

precisely measure the volume of urine we collected, so these possibilities could not be tested in the current study. A controlled study that measures the volume of urine during the daytime can clarify the direct causation of the change in urinary creatinine concentration during day roosting in the active season.

The lack of a sex effect on creatinine concentration during hibernation suggests that there is no significant sex difference in renal function and utilization of torpor (duration and frequency) that influences water reabsorption during winter. This result contrasts with a previous finding in the greater horseshoe bat in South Korea, where a sex difference in urinary creatinine was found during hibernation[28]. This difference may reflect a difference in reproductive physiology between the two species, although we could rule out sampling bias due to the small number of samples in the current study. Ovulation occurs after hibernation in greater horseshoe bats (delayed ovulation), and mating probably continues during hibernation[46,47]. In contrast, ovulation occurs before hibernation in Eastern bent-winged bats (delayed implantation), and thus mating during hibernation is unlikely[48–50]. Therefore, males in the two species may exhibit different arousal frequencies and this can be a possible explanation for such a species difference in urinary creatinine during

hibernation. Although we could not find a sex difference in creatinine concentration during the active season, there was significant sampling bias, particularly in July (1 female urine sample vs 25 male urine samples) and August (5 vs 29). Such sampling bias makes it difficult to rule out a possible sex difference in urinary creatinine during active season which could be influenced by female reproductive events including nursing[51]. Further investigation is necessary to clarify whether such reproductive events can result in a difference in water-stress between sexes.

In this study, we found that the correlation between forearm length and creatinine concentration was weak (Fig. 3d), despite Model 2 (Table 2) suggesting a positive effect of forearm length on creatinine concentration. Forearm length is known to correlate with body mass across bat species[18,52,53], implying that larger bat species likely have a smaller SVR[54,55]. However, inter-individual variations in SVR within a species may be more complex as an individual's body mass changes over time depending on several factors such as pregnancy or feeding[56]. As shown in Fig. 4, Eastern bent-winged bats with heavier body masses produce more diluted urine. This negative effect of body mass on urine concentration changes with the forearm length. We were unable to test whether this interdependence between forearm length and

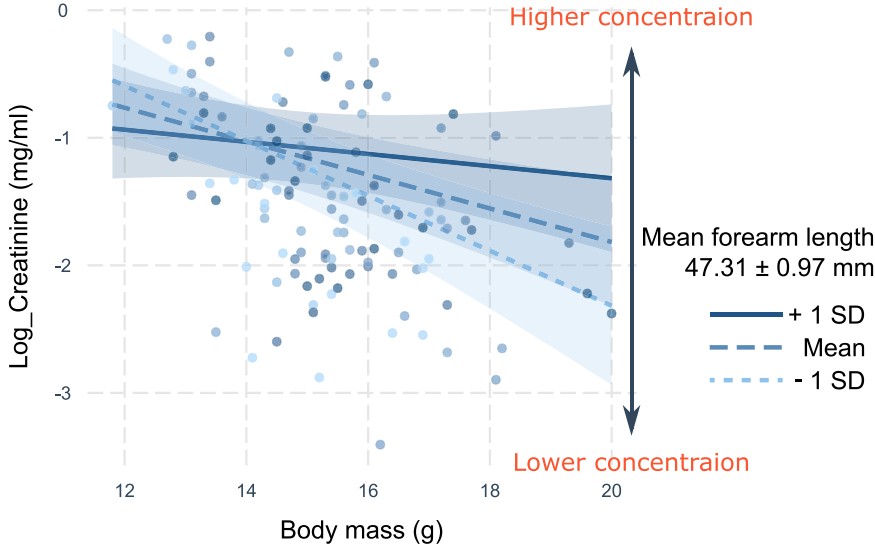

**Fig. 4 Interdependence between body mass and forearm length and their interaction effect on urinary creatinine concentration.** Urinary creatinine concentration decreased with an increase in body mass. However, this negative effect of body mass on urinary creatinine concentration is less pronounced in individuals with longer forearms. The lines are fitted by a generalized linear mixed model (Model 2), and the blue bands around the lines represent 95% confidence intervals. Dots represent individual data points ($N = 119$).

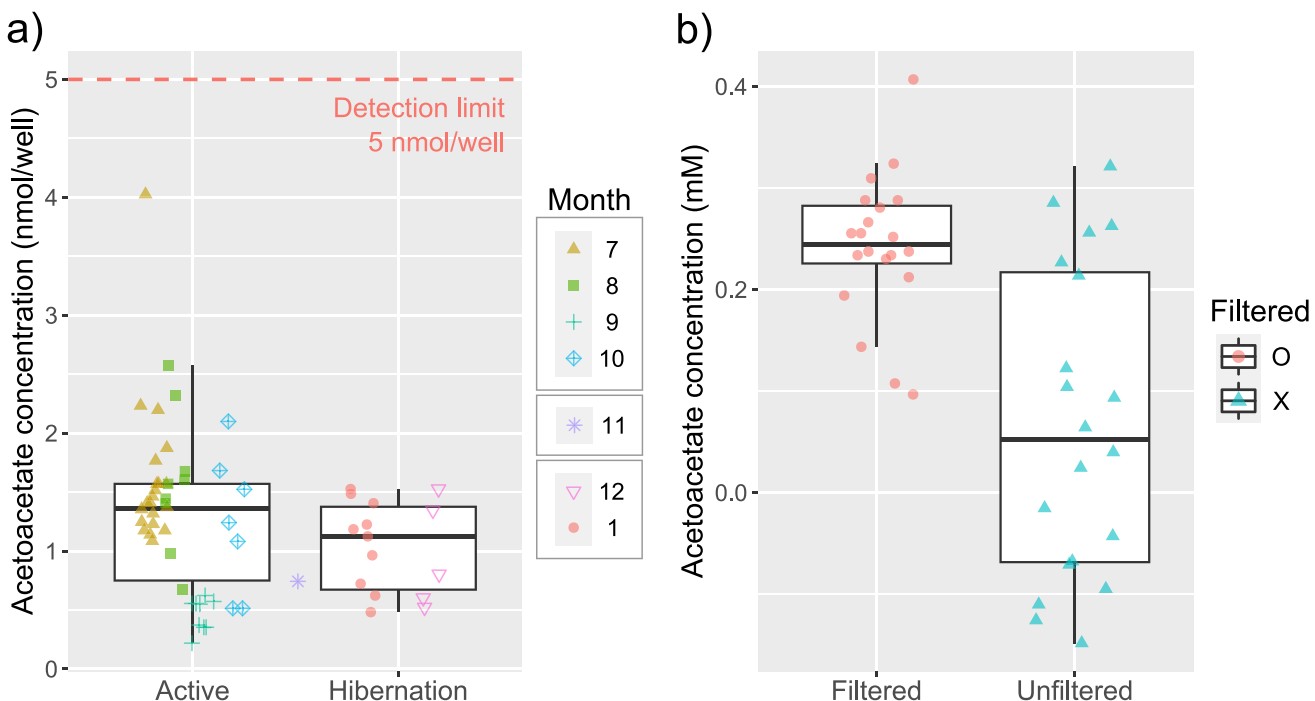

**Fig. 5 Urinary acetoacetate concentration between seasons and between filtered and unfiltered urine. a** None of the urine samples exceeded the detection limit (5 nmol/well) of the assay kit ($N = 62$). **b** Filtered urine samples exhibited less variation in acetoacetate concentration, but their acetoacetate concentration did not surpass the detection limit (20 filtered and 20 unfiltered urine samples). The upper and lower edges of each box represent the 75th and 25th percentiles, respectively. The line within the box indicates the median, the whiskers show 1.5 times the interquartile range, and the symbols represent individual data points.

body mass was due to an increase in SVR in bats with longer forearms. A well-controlled experimental study will help clarify whether water-stress varies depending on an individual's body mass and wing size. Further investigation into this topic will also enhance our understanding of the physiological adaptations of each bat species to water-stress.

We also observed variation in urinary creatinine depending on captured time (feeding status) from July to September (Fig. 1a), indicating that day roosting may increase urine concentration,

while night activities (which may include feeding and drinking) result in lower urine concentration (Fig. 1b). However other night activities such as urination and flight may be influencing factors, particularly in October when prey items decrease. The body mass difference between bats captured at dawn (fed) and dusk (unfed) in each month was consistent from July to September—fed bats were heavier than unfed bats (Fig. 1a and Supplementary Table 1). Therefore, the increase in urinary creatinine from July to September after day roosting was not due to the low amount of

feeding. In addition, as seen in Fig. 1a, b, changes in body mass over months from unfed bats contrast with that in urinary creatinine concentration—the median body mass decreased from July to September, while the median creatinine concentration increased. This contrast between body mass and urinary creatinine suggests that there may be an important physiological process that influences body mass and urine concentration during day roosting from July to September. As seen in Supplementary Fig. 2, the daily TA at the end of the cave was stable and high from July to September. The decrease in body mass after day roosting through to September, therefore, suggests that bats may not enter into torpor[57]. For example, the Eastern bent-winged bats may maintain a longer homeothermic period for better gonadal development during day roosting, as seen in other bats[58,59]. Alternatively, they may switch between roosting locations within the cave that have different RH and TA[60] or change clustering behavior that influences evaporative water loss and metabolic rate that influences metabolic water formation[61]. Unfortunately, we could not further investigate the relationship between the environmental factors (TA and RH) and their physiology due to the malfunction of the data logger. Further physiological and behavioral investigation using thermo- or infrared cameras will be useful to further investigate the monthly changes in urinary creatinine and body mass.

Bats in temperate regions need to balance their energy intake and expenditure for survival and reproduction, depending on the season. Adjustments in energy intake and expenditure are accompanied by physiological processes that leave traceable changes in metabolites in urine. As demonstrated in this study, research on metabolic markers from their urine provides us with a valuable opportunity to infer bat ecology and their physiological adaptation for reproduction and survival during winter[2,62]. Investigations into the physiological ecology of bats in the wild will also contribute to our understanding of the evolution of small mammals and their adaptation to overcome constraints associated with small body size. Furthermore, more accurate and precise tools that require only tiny amounts of samples will contribute to the field of physiological ecology. We hope that more studies and tools will be developed to help us better understand the physiological ecology of small mammals, including bats, in the future.

## Methods
**Study sites and subjects**. Sukkul is a limestone cave approximately 120 m long with a cavernous space around 50 m from the entrance where most bats roost. Hogye mineshaft is around 80 m long, and the floor at the entrance is submerged underwater. Sukkul cave and Hogye mineshaft are about 3 km apart and situated on an edge between a mountainous forest and agricultural fields. Bats dwelling in Sukkul cave and Hogye mineshaft switch their roosting sites from spring to autumn (April to October). There are around 1000 Eastern bent-winged bats in Sukkul year-round, including the hibernation season (December to March). Fewer bats roost in Hogye, and no bats hibernate there except during a short transient period in March.

We captured bats at the entrance of the cave and the mineshaft once a month during the active season using a mist net. We captured bats for 1 h after sunset and for 1 h before sunrise on each sampling night. We started capturing bats 30 min after sunset when they emerged from their roost for night activities. We also started capturing bats 1 h before sunrise in the morning when they began returning to their roost. We adjusted our sampling time according to the time of sunset and sunrise on each sampling date.

After we captured bats, we put them individually into cotton bags (20 × 14 cm; handmade or Ecotone, Poland). After we finished one-hour bat capturing, we started urine sampling as well as banding and measuring body mass (g) and forearm length (mm). We used an electronic scale (RE-260, CAS, South Korea) that is capable of measuring 2 to 500 g with a minimum unit of 0.1 g to measure body mass. For forearm length measurement, we used a digital calipers, CD-15APX (Mitutoyo Corp., Japan) with a minimum unit of 0.01 mm.

We collected urine directly from bats right after we took out them from the cotton bag (Supplementary Fig. 3). When they did not urinate, we carefully patted their abdomen several times. If these attempts failed, we refrained from urine sampling and started measurement of body mass and forearm length and banding. To minimize the disturbance caused by our capturing, we released bats immediately after the sampling and we only conducted sampling once a month at each site. Urine sampling from hibernating bats was conducted immediately after detaching them from a cluster. We then returned them immediately after urine collection without taking body mass or forearm length measurements to reduce any impact on hibernating bats caused by handling and our presence in the hibernaculum. We also did not conduct sample collection in February 2018 to reduce overall disturbance. All urine collection procedures in the hibernaculum took less than 1 h and we have complied with all relevant ethical regulations for animal use.

We recorded TA and RH using EL-USB-2-LCD+ data loggers (Lascar Electronics, UK) in Sukkul. We installed three loggers at the entrance (out), 25 m inside (middle), and 50 m inside (end). We did not install the loggers in Hogye mineshaft because bats do not use the site year-round. To calculate the mean TA and RH of a day, we averaged hourly TAs and RHs of the day from 0 to 24 h. We first defined December to March as the hibernation season and April to November as the active season (non-hibernation season) following previous studies of the activity of bats and moths in Korea[28,63,64]. However, based on our bat sampling records, we excluded November from the active season (see result for further details).

**Urine sample analysis**. We temporarily stored urine samples in an icebox with ice packs at the field site and transferred them to a –20 °C freezer within 4 h of collection. Upon returning to our lab, we stored them in a –80 °C deep-freezer until analysis. The average storage duration in the deep-freezer before urinary analysis was 64.1 ± 36.8 days. Before dilution, we centrifuged urine samples at 6000 rpm (3381rcf) for 1 min (Eppendorf Centrifuge 5424) to remove any suspended materials. We then carefully transferred the urine supernatant to a new microtube. Depending on the amount of urine collected, we diluted urine samples 20 to 33.3-fold with deionized water (Supplementary Data).

We measured urinary acetoacetate and creatinine concentration using commercial urine analysis kits (Abcam, UK): ab204537 for acetoacetate and ab180875 for creatinine. To test whether other urinary components would interfere with acetoacetate concentration, we also tested 20 filtered urine samples using an Abcam 10kD spin column (ab93349). All filtered samples were collected during the active season as the amount of urine collected during hibernation was too small for filtering.

Due to the very low urinary acetoacetate concentration of most of the analyzed urine samples, we discarded any assay results that showed over 50% intra-assay coefficients of variation (% CV). Additionally, we excluded urine samples that produced zero or negative acetoacetate concentrations. We did not set any cut-off values of % CV for creatinine concentration. The mean intra-

assay % CV of the acetoacetate and creatinine analysis was 19.45 ± 12.83% and 2.40 ± 2.13%, respectively.

**Statistics and reproducibility**. We first compared creatinine concentrations between sexes and seasons using a linear model (Model 1). In Model 1, urinary creatinine concentration was the response variable, while sex and season (active vs hibernation) were the explanatory variables. A natural log-transformation was applied to the urinary creatinine concentration to normalize the data distribution. To estimate the potential effect of the micro-environmental difference between the two sites (Sukkul cave and Hogye mineshaft) on urinary creatinine, we included sampling locations as an explanatory variable in Model 1. As only one bat was captured twice during the study period, we did not include individuals as a random effect in Model 1.

Next, we further investigated the effect of body size and feeding status on the creatinine concentration (natural log-transformed) during the active season using a linear mixed model (Model 2). In Model 2, body mass, forearm length, and captured time were included as explanatory variables. Body condition index (BCI) was not included in the model due to recent debates on its reliability and relevance[65]. Instead, we included body mass, forearm length, and their interaction in the model.

Forearm length is known to correlate well with body length[66]. Additionally, a longer forearm results in a larger wing area, and their body mass is also correlated with wing size across bat species[18,52,53]. Therefore, it is likely that bats with heavier body mass have smaller SVR across species. However, within a species, there could be inter- and intra-individual variations of SVR depending not only on the forearm length but also on an individual's body mass that changes in relation to feeding or reproductive status. For example, bats with longer forearms that starved for a day may have greater SVR due to a decrease in body mass than those that have shorter forearms but greater body mass. In this case, the increase in SVR of the bats with longer forearms but lighter body mass can result in higher evaporative water loss from their skin. This possibility is examined in the model by the interaction term between body mass and forearm length.

To avoid the high variance inflation factor (VIF) caused by the interaction term, we Z-transformed the body mass and forearm length in Model 2 by subtracting the mean from each value and then dividing them by the standard deviation: $(x - \text{mean})/\text{sd}$. The feeding status of the bat was determined by the timing of urine collection. Bats that were caught at dusk when they emerged from a cave were considered unfed. When bats returned to a cave after night activity at dawn 1 h before sunrise, they were considered fed. To estimate the effect of sex on creatinine concentration, we also included sex as an explanatory variable. Sampling locations (Sukkul or Hogye) were included as a random variable in the model to control for the effects of possible microenvironmental differences, even though Model 1 revealed no significant effect of sampling location. After running each model, we used the "summary" function in R to investigate the model coefficients. Confidence intervals were calculated using "confint" function with 1000 times bootstrapping.

Finally, we further investigated the interaction term between body mass and forearm length in Model 2 using an R Package called "interactions[44]". With the package, we conducted simple slopes analysis and plotted the effect of body mass on urinary creatine depending on three forearm lengths (mean and mean ± 1 SD).

We used R 4.3.0[67] and several packages including "lme4[68]", "lmerTest[69]", "car[70]", "ggplot2[71]", and "interactions[44]" for our statistical analysis and graphics. We verified the residual

distributions using statistical tests (Shapiro–Wilk) and visual inspections, such as Q–Q plots, to ensure there were no violations of the model assumptions. Throughout the inspection, we did not find any violations of the assumptions. To estimate any multi-collinearity between explanatory variables, we calculated the VIF. The maximum VIF was 1.34, indicating no serious multi-collinearity between the explanatory variables, including the interaction term in the model.

**Reporting summary**. Further information on research design is available in the Nature Portfolio Reporting Summary linked to this article.

## Data availability
The raw data[72] collected and analyzed during the current study are available at the Figshare online repository (https://doi.org/10.6084/m9.figshare.21197350).

## Code availability
The codes for the statistical analysis in the current study are available from the corresponding author on request.

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

## Acknowledgements

We thank Somi Yoo, Injung Ahn, Byeori Kim, and Hyeongcheol Park of NIE for their research contributions to research management and assistance with urine analysis. We also thank Ryeon Kim, Yuri Kawaguchi, and David A. Hill for their valuable comments and English corrections. Lastly, we would like to express our appreciation to the two reviewers and the editor who provided invaluable input and suggestions for this paper. Their contributions have significantly enhanced the quality of our work. The National Institute of Ecology in South Korea funded the current project (NIE-C-2017-02, NIE-C-2018-02, NIE-C-2019-02, NIE-C-2022-02 to Kim). The funding body had no role in the design of the study, data collection, data and sample analysis, interpretation of the data, or writing the manuscript. The internal research evaluation committee, consisting of three external investigators, evaluated the project at the end of each year.

## Author contributions

All authors have read and approved the manuscript. H.R., K.K., S.K. conceived and designed the study. H.R., K.K. performed the experiments, H.R., S.J., Y.C., S.K. collected samples. H.R. original writing, H.R., K.K., Y.C., S.K. revision and editing.

## Competing interests

The authors declare no competing interests.

## Ethics approval

The current study was conducted as a part of the Basic Ecological Study Programs (1105-1 to Kim). This study was approved by the Research Planning Review Committee of the National Institute of Ecology. We declare that we adhered to the Wildlife Protection and Management Act of Korea and Institutional Research Ethics Regulations and Guidelines. All handling and sampling permission was approved by the local government (Mungyeong 2017-1 to Kim). We have complied with all relevant ethical regulations for animal use.
