## [Peer Review File · Communications Biology]

Reviewers' comments:

Reviewer #1 (Remarks to the Author):

In their paper "No evidence of ketonuria, but seasonal variations in urinary creatinine were found in the Eastern bent-winged bat (*Miniopterus fuliginosus*) in Korea" Ryu et al investigate the potential role of evaporative water loss in bats during the active vs hibernation season. They have undertaken this study using novel methods one of which seems to have been insufficient to produce viable results while the other has yielded some interesting findings. I think there is great potential in this paper to provide some new information about an understudied topic, especially as it concerns measuring animals within natural roosting sites over the course of the year. However the presentation of the paper was generally confusing and lacks a clear flow of logic beyond the introduction. Although the results and discussion are separated into subheadings the links that the authors make between their findings and conclusions is not always easy to follow. They also seem to come to some circular conclusions. One important facet is their heavy focus on the relationship between body mass and forearm length. This entire presentation and the discussion surrounding it feels like the authors are trying to explain correlations that exist as if they are causative. The variation in body mass in the individuals sampled is only ~4g and varied with the timing of sampling. Most of their discussion around the potential relationship between this and evaporative water loss is highly speculative.

The paper would benefit from an extensive rewriting to ensure that the logical flow is more clear and that the authors speculations are limited and the discussion more well connected to their findings. It is essential that the details concerning their evaluation of acetoacetate is clarified and if, as it appears to state in the results, the method did not produce viable results, then the title, abstract and discussion must be reframed to reflect only the results for creatinine concentrations.

I have provided additional comments throughout the manuscript file. (Editorial Note: This file should be attached to this email with comments/suggestions from Reviewer #1. If you or your co-authors have any difficulty downloading the attachment or understanding the comments/suggestions, please feel free to reach out to the editorial team at Communications Biology for clarifications or help).

Reviewer #2 (Remarks to the Author):

In this study, the authors measured and evaluated urinary metabolic wastes (creatinine and acetoacetate) in the Eastern bent-wing bat (*Miniopterus fuliginosus*) captured at a limestone cave and a mineshaft in South Korea. These sites were used by the bat population from spring to autumn and one of these sites also during the hibernation season. While there was no evidence of increased excretion of urinary ketone bodies during hibernation, seasonal variations in urinary creatinine indicated more concentrated urine production during hibernation. The interpretation and discussion of obtained results may provide some insights into the metabolic switch in utilization of glucose, proteins, and lipids and renal function in terms of glomerular filtration, reabsorption and urine formation preventing metabolite and water loss in insectivorous bats of the temperate zone in relation to their annual cycle of activity versus torpor and arousal bouts. Importantly, bats were sampled non-invasively and non-lethally.

The manuscript is rather well written in a readable way and can be recommended for publication following some improvements as suggested in following comments.

Comments:

- 1) Please provide argumentation for the selection of the bat species for the present study.
- 2) Please provide data about collected urine volumes (only the number of urine samples mentioned in Table S1).
- 3) On lines 370-371 you should be more specific about the urine collection procedure. In my opinion, the term genitalia is not correct. Wording of the sentence may also be understood as collection of urine after taking genitalia out of a cotton bag.
- 4) Tools and their accuracy to measure body weight and forearm length are not mentioned.

Our revisions and responses to the reviewers are as follows.

We made some minor but important corrections to grammar and contextually inappropriate usage that was not pointed out by reviewers and the editor. These minor corrections were also highlighted (yellow shade) in the manuscript with our revision that was asked by reviewers. We also found two important mistakes in the previous manuscript. The first one was that we made a mistake in the calculation of the detection limit of the acetoacetate kit. The detection limit of the kit was 5 nmol per well (not the 2.5 nmol per well that is previously reported). We found this while we were discussing the acetoacetate kit validation with the manufacturing company, called Abcam. We revised it (L 219) and based on this we also revised Fig. 5 (previously Fig. 3). We also discovered two missing records for the sex of the two bats whose urine samples had been labeled as unknown sex. Therefore, we ran one statistical analysis (Model 1) again and updated its result in Table 2. We also updated Table 1, and Fig 2 respectively following the changes in data. Fortunately, the addition of the two samples to the dataset neither changed any statistical significance nor resulted in any noticeable changes in statistical tendencies.

Response to Reviewer #1 (general comments)

Your comments

In their paper "No evidence of ketonuria, but seasonal variations in urinary creatinine were found in the Eastern bent-winged bat (*Miniopterus fuliginosus*) in Korea" Ryu et al investigate the potential role of evaporative water loss in bats during the active vs hibernation season. They have undertaken this study using novel methods one of which seems to have been insufficient to produce viable results while the other has yielded some interesting findings. I think there is great potential in this paper to provide some new information about an understudied topic, especially as it concerns measuring animals within natural roosting sites over the course of the year. However the presentation of the paper was generally confusing and lacks a clear flow of logic beyond the introduction. Although the results and discussion are separated into subheadings the links that the authors make between their findings and conclusions is not always easy to follow. They also seem to come to some circular conclusions. One important facet is their heavy focus on the relationship between body mass and forearm length. This entire presentation and the discussion surrounding it feels like the authors are trying to explain correlations that exist as if they are causative. The variation in body mass in the individuals sampled is only ~4g and varied with the timing of sampling. Most of their discussion around the potential relationship between this and evaporative water loss is highly speculative.

Our response

First of all, we would like to appreciate your comments and guidance. Your comments were very critical and helpful for us to realize that some parts of the discussion could be wrong. Particularly, we realized that our interpretation of the effect of body weight and forearm length

on creatinine concentration could be wrong (See also our response to you in the discussion, LL 253 to 264). We focused much on the interdependency between body weight and forearm length and their effect on urinary creatinine. As you pointed out the forearm length could be a simple correlation that has no causative effect on the evaporative water loss (EWL) given the high humidity in the cave. It is also unlikely that the forearm length would influence the breath rates. As we agreed with your point, we tone-downed and removed much discussion about the effect of forearm length on urinary creatinine concentration in the result and discussion parts.

We previously thought that EWL will be influenced by forearm length as it will increase the surface area. In addition, we assumed that the heavier bats would lose more weight during day roosting. As shown in the figure below (previously Fig. 6a, but removed in the revised

document), forearm length and body weight positively correlated in fed bats. However such correlation has gone in unfed bats. From this relationship, we thought that bats with longer forearms and heavier body weight would lose more

weight during day roosting. However, as you pointed out, we cannot be 100% sure about it as we did not track the weight change from an individual bat during day roosting. We removed all related parts based on this possibly wrong assumption. We hope that our revision will meet your expectations.

We also revised our discussion on the possible interaction effect of forearm length and body weight on the variations in creatinine concentration. As you pointed out it is uncertain whether there would be a causal relationship between creatinine and forearm length only with our data. We tone downed our discussion by removing much discussion on this topic and pointed out what the limitation of our data from the field was. We also agreed with your point that the discussion surrounding the relationship between up to 4 grams of variations in body weight and evaporative water loss was speculative. However, 4 grams are over 20% of their body weight and bats increased about 10% of their body weight on average after feeding in our data. These large changes in body weights could have some effect on SVR as you also pointed out in the discussion (LL 288-289). To raise some questions and provide a possible direction for a more controlled study in the future, we left some of our suggestions in the discussion. We hope that our revision will meet your expectations.

Your comments

The paper would benefit from an extensive rewriting to ensure that the logical flow is more clear and that the authors speculations are limited and the discussion more well connected to their findings. It is essential that the details concerning their evaluation of acetoacetate is clarified and if, as it appears to state in the results, the method did not produce viable results, then the title, abstract and discussion must be reframed to reflect only the results for creatinine concentrations. I have provided additional comments throughout the manuscript file.

Our response

We moved tables and figures to the main manuscript following your suggestions and did an extensive rewriting of the manuscript particularly in the result and discussion parts to integrate your concerns. In particular, we removed most of the acetoacetate results and moved its position to the end of the result part (See LL 216 to 222 for revision) – it was previously located at the beginning of the result. We previously thought that based on the very linear relationship between the acetoacetate concentration and the optical density (absorbance) demonstrated in the kit manual, even a very low concentration can be a meaningful result if the experiment was very precisely conducted. However, as you suggested, we could not rule out the possibility that our assay results were just derived from the high dilution rate, rather than resulted from the low concentration of acetoacetate in urine. In addition, we also found a mistake in our calculation of the kit detection limit. We deleted most of our discussion about the meaning of low acetoacetate concentration and its implications. We also removed most of our discussion on urinary acetoacetate concentration as it could be misleading and we deleted much discussion about the effect of forearm length that was pointed out as speculative. We also changed the title and abstract as you suggested following our revision. We hope that our revision and responses will meet your expectations.

Response to specific comment to Reviewer #1

Thank you for your comments on the manuscript. We accepted your changes on the manuscripts mostly and left track changes on the manuscript. As we mentioned before, we realized that our discussion about the effect of forearm length had a hidden assumption that it would influence the SVR so it would affect EWL. However, as you pointed out the effect of forearm length can be very small, particularly in a cave where relative humidity is high. We revised it to include the points that you raised as much as possible.

Here comes our response to your specific comments that we copied from the manuscript. We also left our comments directly on the manuscript so please use this response letter (preferred) or the manuscript for your convenience. Due to our revision, some of your comments have no line numbers as they were deleted in the revised version. However, we made responses to all deleted comments as well. To see the deleted comments, please check the attached manuscript that we

left track changes for you. We left our responses there – you can see them by selecting Showing all markup in the Review pane in the Word menu. The line numbers we referred to here are from the revised document, not the original draft we submitted. We hope that our revision will meet your expectations.

Your comments (LL 41-43)

I'm not sure I grasp your reasoning here. It depends on your definition of small mammals, but heterothermic species tend to be quite small compared to homeothermic species. Do you mean amongst heterotherms, hibernation is challenging for smaller species? Because being small and having the ability to lose heat quickly enables flexible use of torpor and massive energy savings- but its kind of a chicken and egg discussion.

The savings are huge when animal enter torpor, so I don't really see how high EWL is an issue for small hibernators, especially cutaneous EWL when these animals are known to reduce peripheral blood flow to basically zero. The way this sentence is currently constructed it reads as if torpor increases EWL in small species....

I would suggest rewording this entirely.

Our response and changes

In the original draft, we wanted to point out that small mammals might have higher EWL during hibernation due to their SVR. As you pointed out, however, our logical flow had some circular statements and was unclear to express what we would like to suggest. We decided to remove most of our arguments on EWL and fat storage in small mammals and rewrite this part to talk about bats. The original sentences and the revised sentences are as follows (LL 41-43). Red and underlined letters represent the revised parts from the previous sentences.

Original: "However, as their small body size limits fat storage and increases heat and water loss from the skin due to their greater surface-to-volume ratio (SVR) compared with large mammals, hibernation is also more challenging for small mammals^{3,6,7}."

Revised: "For bats, in particular, due to their small body size and greater surface-to-volume ratio (SVR) compared with other small mammals, hibernation can be challenging^{6,7}."

Your comments (LL 45 previously but deleted due to revision)

I don't see the logical link to the sentence above. If hibernation (which by definition is a reduction in Tb) is difficult for small mammals, then why is it now adaptive?

Our response and changes

There might be a misuse of ‘therefore’ here. We would like to say that hypothermia is an important adaptation to reduce water and energy loss during hibernation. The previous sentence was to emphasize the difficulty of hibernation of the small mammal due to its limited storing capacity so it had no logical link in between. We revised sentences (LL 43-45) from:

“Hypothermia during hibernation is, therefore, a key adaptation to energy and water-saving and thus winter survival for small mammals⁸.”

to

“Hypothermia in bats that allows reducing metabolism and water loss during hibernation may be an important adaptation to energy and water-saving and thus winter survival⁸.”

Your comments (LL 61-62)

Is there evidence from species other than bats?

Our response and changes

Other species may use different strategies, so we limited this to bats and revised the paragraph from LL 57 to 66 to integrate your points.

Your comments (LL 72-73)

However the skin of bat wings are embedded with lipids to reduce cutaneous EWL compared to other mammal species.

Our response and changes

We revised it by adding your point and a reference. Revised sentences are as follows (LL 70-73).

Although the presence of lipids in bat wings may prevent much evaporative water loss¹⁸, the large SVR of bats also increases evaporative water loss from the skin making it difficult for bats to maintain body water balance⁸.

Your comments (LL 74-75)

Not all, you are ignoring the great array of insectivorous temperate zone bat that hibernate in tree hollows.

Our response and changes

Oh, yes. Thank you. Revised from:

“small insectivorous bats” to “many small insectivorous bats”

Your comments (L 141)

Did you measure body weight or body mass? Weight requires corrections for gravity.

Our response and changes

Our measurement was probably weight. We used a weighing scale that we calibrate with a load with a fixed mass that was provided with the scale. Let us know if this should be corrected to mass.

Your comments (L 147)

How many bats total and of what sex?

Our response and changes

Thank you very much for your asking. We integrated the information you asked for (L 146). To answer your question, we checked two different notes that were recorded by another researcher who was there at the same time. With this double-check, we could determine the sex of the two sex-unknown urine samples in the previous manuscript. As a result, the total number of samples in Table 1 (previously Supplementary Table 1) changed from 160 to 162. Fortunately, no results of Model 2 about the relationship between body weight and forearm length on the creatinine in Table 2 (previously Table 1) and Supplementary Table 1 (previously Supplementary Table 2) have changed. It was because the two newly sex-determined samples were collected during winter when we did not measure the body weight and forearm length of the individuals. We updated this information with all necessary parts – Table 1, Table 2 (the part that represents the result of Model 1), and Fig 2 were updated, and other figures stayed the same with the previous version.

Your comments (L 150)

How were these partitioned for analysis and do you have any data on the volume of the samples?

I realize now that some of this data is in the supplementary materials however the details of where these were collected is not available.

Our response and changes

We wonder about the meaning of the partition of the data. Did you mean how much urine we used for each urine analysis? We followed the protocol provided by the kit from Abcam. For a

double test of one sample, we needed 320µl of diluted urine at least (100µl for creatinine and 220 µl for acetoacetate). As usual urine volume collected was about 20µl, we did 20x dilution mostly. However, some samples also had different dilution rates because they were less than 20µl. This information is all provided in the raw data that we uploaded at Figshare (See data availability part). We could not measure the volume of the whole urine samples we collected as it was not possible to exactly measure the volume as we previously mentioned. We also freeze them as soon as possible after we collected them. After we thawed the sample for the hormonal analysis, we took the exact necessary amount of them for dilution e.g., for 20x dilution we took 5µl of urine and diluted it with 95µl of ddH₂O for creatinine. We included the number of samples from the different caves and moved Table 1 to provide more details in the main manuscript.

To include what you asked, we revised the sentence from

“In total, we collected 162 urine samples (Table S1).”

to

“In total, we collected 162 urine samples (Hogye: 30, Sukkul: 132, Table 1). There was only one resampled bat that captured once in October (active season) and December (hibernation season).”

Your comments (LL 154-164)

The choice of what data is presented in the body of the paper and what is left to the supplementary materials seems strange to me. I would prefer to see the figures and tables of the supplementaries than a map of the study site and long presentation about Ta and RH across the cave when there was no information regarding where these bats might be roosting in the cave- not even for the hibernating bats who were taken directly from their hibernation cluster. It would be better to give us this data in a relevant context. Also- why is the RH never used in any of the analyses?

Our response and changes

Thank you very much for your suggestion. We agreed with your suggestion. Following your comment, we moved Fig. 1 and 2 to the supplemental information (Supplementary Fig. 1 and 2) and, Supplementary Table 1, and Supplementary Fig. 1 to the main manuscript (Table 1 and Fig. 1 respectively). As we provided raw data, we thought that such details can be easily calculated who want more details. However, as you suggested, it would be easier for readers to see them if they were presented in the main manuscript. Table S2 became Table S1 and stayed in the supplementary data as some of these data are presented in Fig. 1 (previously Supplementary Fig. 1). We updated these changes throughout the whole document. We also moved the result of acetoacetate behind the creatinine result following your suggestion.

We were not able to collect their roosting locations in the cave during the active season. When we entered the cave, bats usually woke up and flew. We saw this when we installed temperature loggers. However, from our experience and given the cave structure, we guess that bats most of the time, roost at the end of the cave even during active periods. During hibernation periods, we could only find bats at the end of the cave. That information was shortly mentioned in the method previously. We were not able to confirm whether the logger installed at the end of the cave worked well or not - the RH dropped suddenly for a few months and increased suddenly. We contacted the company, but they also told us that they could not be sure whether it worked properly or not. That was the reason why we did not use the RH data for further analysis.

Your comments (L 167)

Again- weight or mass?

Our response and changes

We used weight. Let us know if we need to change weight to mass. As far as we know, mass can be measured with a balance, and weight can be measured by a scale. We used a scale that was calibrated regularly with a known (certified) mass.

Your comments (LL 168-169)

But there does appear to be an interesting trend of reducing BM in the previous months amongst the unfed bats. And the BM of unfed bats in October appear to be higher than previous months.

Our response and changes

Yes, exactly. Thank you for pointing this out. To integrate your suggestion, we revised our sentences as follows (LL168-172):

“Nevertheless, bats captured at dawn were not always heavier than bats captured at dusk as in October (Supplementary Table 1; Fig. 1a). This result might be because feeding success was low in October due to fewer prey insects. Although there was a decreasing trend in the body weight of unfed bats until September, this trend reversed in October.”

Your comments (L 186)

Why are there no degrees of freedom in the table?

Our response and changes

R summary function does not provide df for each independent variable. We, therefore, added it by adding the model equation of Model 1 at line 184. You can find the DF (158) from the f-statistics. For Model 2, we included all DFs from the model summary function in R.

Your comments (L 189)

How many samples were collected in each location?

Our response and changes

For Hogle, there were 30 urine samples and for Sukkul there were 132 urine samples. We added this to the previous page (L 150). Shapiro–Wilk test suggested that the urinary creatinine concentration (log-transformed) of the Hogle samples might not distribute normally. However, the two-sample Kolmogorov-Smirnov Test showed that there's not enough evidence to say that those samples came from two different distributions ($D = 0.17727$, $p\text{-value} = 0.3779$, two-tailed).

Visual inspection (left figure) of the two groups of urine samples using QQplot also suggested that all values were within 95% CI. Therefore, it would not be wrong to suggest that no difference between the two sampling locations.

Your comments (L 198)

Is this not potentially confounded by whether or not they had fed?

Our response and changes

Yes, feeding might be a confounding factor that could influence the effect of weight on urinary creatinine. As shown in Fig. 3b, for the unfed bats, body weight was an important determining factor of the creatinine concentration. However, for the fed bats, the creatinine concentration was not related to body weight. This result suggests that feeding may lower the creatinine concentration in general.

Your comments (LL 201-203)

OK I see you have tried to deal with this here.

Our response and changes

Yes, indeed. Unfed bats with lighter weights showed higher creatinine concentration. This is strange for us and the reason why we discussed a lot from this relationship. However, as you suggested, this could be a simple correlation, so we removed most of our discussion about this and tone-downed the point that we suggested.

Your comments (LL 203-204)

These two sentences are repetitive consider rewording to make it more concise.

Our response and changes

We removed the other part of the sentence to make it clear. We hope that this revision made it more concise. We revised sentences (LL 203-204) from:

“There was no sex effect on urinary creatinine concentration, which is consistent with the result in Model 1. As shown in Fig. 5c, the negative effect of body weight on creatinine concentration was consistent between the sexes.”

to

“Urinary creatinine concentrations did not differ between sexes (Fig. 3c).”

Your comments (LL 204-205)

These two sentences are repetitive consider rewording to make it more concise.

Our response and changes

We revised sentences (LL 204-205) from:

“Forearm length was positively related to urinary creatinine concentration although such a positive relationship was inconsistent in September (Fig. 5d).”

to

“Bats with longer forearms exhibited higher urinary creatinine concentration over months except those who captured in September (Fig. 3d).”

Your comments (LL 208-215)

This entire section is extremely confusing and the logic behind the analyses is not clear to me. Is there some a priori reason to believe that forearm length would ever correlate with creatinine concentrations? Considering that bats in this study varied and we know nothing about the actually feeding status of a given individual (there is no comment about whether or not individuals were recaptured at any point in the study) I feel like this is a strange road to take and is kind of just exploring any possible correlation.

Our response and changes

Thank you very much for your point. After we finished our revision, we got to understand your concerns better. We now agree with your point that this could be a simple correlation. We removed Fig 4a (previously Fig 6a and found on the 3rd page in the response letter) and the related analysis to the Figure.

Before we made responses to your concerns, we would like to clarify the number of samples and recapturing. In Model 2, we used all urine samples (133 urine samples) collected from July to October from different bats. There was no recapturing – all bats were captured only once in this season. The only re-captured bat was captured once in October and once in December, so recapturing cannot influence the results of Model 2 and the following analysis. We added this information at the beginning of the result in this revision (LL 150-152).

As you pointed out, the relationship between the forearm and creatinine concentration could be a simple correlation. However, the reason why we added the forearm length in the model was we thought that it would influence wing size and wing surface area which can finally influence the SVR. In addition, this part was a further investigation of the significant interaction term between body weight and forearm length on urinary creatinine, rather than an investigation of the effect of forearm length only. As the interaction was significant, it was necessary to be investigated to understand the main effect better. If your question was why we added this interaction term in the model, we would say we thought that there could be some inter-dependency between forearm length and body weight on urinary creatinine. Our logic in more detail is as follows.

When forearm length increases, wing area size will increase more than its length as it can increase exponentially (2-squared in maximum). For a more detailed discussion, let's think about two bats with the same body weight but different forearm lengths. This difference will result in different SVRs in the two bats. In this case, the bat with a longer forearm probably has a larger surface area than the bat with a shorter forearm as their body weight is the same. We know that this could be a too simple assumption. We could not find a good reference to support this assumption either. However, as volume increases more rapidly than its surface area, when body weight is the same, the bat with a shorter forearm probably has a larger volume (and smaller SVR) than the bat with a longer forearm. You also pointed this out in the discussion (LL 288-289).

We did not integrate this assumption (logic) into the introduction as we did not consider such interdependency between the body weight and forearm length on urinary creatinine when we designed the study. By not including this logic in the introduction, we could also avoid recent concerns about p-hacking and false positive findings. We started to understand the interdependency between forearm length and body weight on urinary creatinine concentration after we further investigated the interaction term. This is the reason why we added it in the result part and discussed it further later in the discussion part.

Again, when we ran the statistical analysis, we started to think about the interaction between the two variables but did not predict any interdependency. However, after we ran the statistical test, we found this significant interaction. We understand that the reader will feel strange about this part and may think that this is too sudden as you pointed out. However, as this interaction may suggest some interesting conditional effects of forearm length and body weight on urine concentration that can help future studies, we tried to integrate this part. When we prepared this manuscript, we thought about integrating this part only into the discussion. However, as new results and analysis in the discussion are generally not recommended, we put this analysis in the result section. However, as some extended analysis of the result is allowed in discussion for this journal, we can move this part to the discussion or consider removing it if it does not make any sense. Let us know what you think.

Original sentences and revised sentences are as follows:

Original

“To better understand the effect of the interaction term, we conducted a further investigation on the significant explanatory variables in Model 2. First, we investigated whether there was a relationship between forearm length and body weight depending on feeding status. As shown in Fig 6a, bats with longer forearms were heavier than those who with shorter forearms when they came back to roost at dawn after feeding: significant interaction between feeding status and forearm length (linear model $R^2 = 0.244$, $F(3, 115) = 12.39$, $p < 0.001$). Second, we investigated how the effect of body weight changed depending on forearm length during daily roosting and night activities in Model 2 by fitting the effect of body weight with three forearm lengths (mean forearm length and mean \pm 1SD). As a result, we found that heavier bats produced lower creatinine-concentrated urine during daily roosting. This negative effect of body weight on creatinine concentration was stronger for the bats with shorter forearms – heavier bats with shorter forearms had significantly lower urinary creatinine concentration (Fig. 6b).”

Revision

“To clarify this dependence between the two variables we investigated how the effect of body weight changed depending on forearm length during daily roosting and night activities in Model 2 by fitting the effect of body weight with three forearm lengths

(mean forearm length and mean \pm 1SD). As a result, we found that heavier bats produced lower creatinine-concentrated urine during daily roosting. This negative effect of body weight on creatinine concentration was stronger for the bats with shorter forearms – heavier bats with shorter forearms had significantly lower urinary creatinine concentration (Fig. 4a).”

Your comments (L 218)

This abbreviation needs to be spelled out.

Our response and changes

As this section moved to after the creatinine result, we added the full spell of CV (coefficient of variation) in the previous section. Thank you.

Your comments (LL 226-227)

I thought you couldn't find anything with this data therefore any mention of hibernation in this context is misleading. It should say that the method was unable to accurately detect urinary acetoacetate at all.

Our response and changes

Good point. We decided to be more conservative in our data interpretation following your suggestions and concerns. We removed most of our discussion about the acetoacetate. One thing we would like to point out is that the colorimetric method used for acetoacetate, results in a linear relationship between the acetoacetate concentration and optical density (OD) absorbance. Therefore, if we conducted the experiments very accurately – this can be evaluated by the CV – the low OD values in the analysis can represent the true but very low acetoacetate concentration. However, this reasoning and argument are not the purposes of the paper, so we decided to remove such possible interpretations. We hope that our revision will meet your expectations. Our revision and the original sentences are as follows.

Original

“Although we could not find any evidence of an increase of urinary acetoacetate (ketone bodies) during hibernation, we did find seasonal variations in urinary creatinine that indicate more concentrated urine production during hibernation.”

Revised

“We could not find any valid acetoacetate (ketone bodies) concentrations from urine probably due to our high dilution rates. However, we did find seasonal variations in urinary creatinine that indicate more concentrated urine production during hibernation.”

Your comments (L 230)

The way this is written makes it sound like you think torpor itself is causing evaporative water loss and this is really misleading. As I said earlier EWL in torpor is substantially reduced compared to active animals so the reasoning is more that they do not intake additional water when torpid, not that prolonged torpor increasing evaporative water loss.

Our response and changes

We removed the EWL part following your suggestions and accepted your corrections. However, the sentence seems to be less clear. The limited water intake due to prolonged torpor during hibernation may not result in concentrated urine unless there was body water loss caused by urination or EWL. We cannot think of other activities that can increase water reabsorption throughout glomerular filtration that increases urine concentration during hibernation. The original sentence could be misled as you pointed out if it could be read as prolonged torpor will increase evaporative water loss. However, our point was that even with lower EWL, if they did not drink for a long time, the EWL (including loss from breathing) could be the main reason for body water loss unless they urinate much, so need to increase water reabsorption by glomerular filtration to compensate their body water loss through urination. Original sentences and revised sentences are as follows:

Original

“This finding suggests that Eastern bent-winged bats produce concentrated urine due to limited water intake or evaporative water loss from prolonged torpor bouts during hibernation.”

Revised

“This finding suggests that Eastern bent-winged bats produce concentrated urine due to limited water intake that is related to prolonged torpor bouts during hibernation.”

Your comments (LL 233-246)

This needs clarification for me- obviously I don't have a background in this method but from how the results are constructed it appears that you are unable to make any conclusions from your results. Either make this more clear or remove these details and speculations from the discussion.

Our response and changes

We removed most parts of this discussion and rewrote all after line 236 to include the possible reasons of no detection of urinary acetoacetate. As we explained before, the colorimetric method that was used to measure the acetoacetate concentration in this study produce a linear relationship between the concentration and OD (optical density) absorbance. So, if the CV was not too bad, we could still say that the acetoacetate concentration was low. There is a way to test this possibility. As the lowest limit of the kit was 5nmol per well, we can spike all our samples by adding 5nmol acetoacetate and reanalyze all samples if we have more samples. We don't think that such a method will show different outcomes from the results we got previously. Of course, to do that we need to run further validation tests (such as recovery tests) to confirm the reliability of the spiking test. However, we could not do that as we don't have enough left-over samples as we spent almost all urine samples even though we applied x20 dilution mostly. It is unfortunate that we could not convince you, but we would like to leave some clues for the next researchers who want to test this possibility for their study if you agree with our revision. Original sentences and revised sentences are as follows:

Original

“Urinary acetoacetate and arousal during hibernation

Serum accumulation of ketone bodies from intensive fat metabolism during hibernation will decrease serum pH. To balance the serum pH level, hibernating bats may arouse to transfer ketone bodies from serum to urine⁴³. However, we could not find any evidence of a significant increase in urinary acetoacetate concentration. Instead, we found the highest urinary acetoacetate from bats in the active season. During the coldest time in winter, bats enter the deepest torpor, which reduces metabolism and arousal frequency significantly. Reduced metabolism during hibernation will then decrease the formation and accumulation of ketone bodies from fat metabolism. In this case, hibernating bats will not face difficulties in maintaining serum pH and will not arouse to discard serum accumulation of ketone bodies in urine. In addition, given the limited energy budget of hibernating small insectivorous bats, ketonuria caused by an increase of serum ketone bodies, including acetoacetate from intensive fat metabolism, would not be adaptive for hibernating bats.”

Revised

“Serum accumulation of ketone bodies from intensive fat metabolism during hibernation will decrease serum pH. To balance the serum pH level, hibernating bats may arouse to transfer ketone bodies from serum to urine⁴⁵. Although the current study aimed to clarify this possibility, we could not confirm it as our urinary acetoacetate analysis did not produce meaningful results. There could be two possible explanations for our result. One is the high dilution rate (at least 20 times) due to the limited amount of urine samples made it impossible to measure the urinary acetoacetate concentration. As the detection limit (sensitivity) of the kit was 5nmol per well, at least a 20x dilution rate in the current

study made it impossible to detect acetoacetate even if there was enough acetoacetate in undiluted urine samples. The other possibility is that the Eastern bent-winged bats in the current study did not exhibit ketonuria during hibernation and active season. This possibility can be tested using other methods including paper strip test with undiluted urine samples or mixing up several urine samples collected in the same season to make enough volume for urine analysis.”

Your comments (LL 255-266)

I think this line of reasoning is backwards. As the seasons progress and ambient temperature falls the bats are likely to use MORE torpor. The increased filtration when bats are using less torpor might result in lower creatine concentrations because of increased urination throughout the day as bats often urinate right after arousal. For this reason some indication of the volumes of urine sampled would be very interesting to see. In addition this reasoning may explain lower creatinine concentrations in the urine of the “fed” bats collected in the morning.

Our response and changes

Thank you very much for your point that made us realize the deficiency in our logical flow. We had a hidden assumption that bats with heavier body weights may increase their arousal time for the development of their germ cells. However, we do not have any evidence of arousal time data to support this argument. In addition, we don't have any data to support our assumption that such longer arousal can increase urinary creatinine unless those bats exhibited longer arousal time urinated and defecated more and loose body water that finally would increase water reabsorption and urinary creatinine concentration. This logical flow needs much speculation and is probably not the case. We revised this part heavily and removed most of the discussion about the significant interaction term between body weight and forearm length. Thanks a lot again for your help.

We now understand what made you very concerned about our presentation and our interpretation of the creatine result. We agreed with your point that the lower concentration in the heavier bats after day roosting could simply reflect that they might produce more urine of which creatinine concentration was low. It could be a simple correlation that indicates larger bats (bats with longer forearms) are just heavier due to their larger body size. If this is the case, it is a simple correlation and much discussion based on that does not make much sense. We focused too much on the significant interaction and tried to fit our discussion to explain the significant interaction even though it could be a simple correlation.

In the previous draft, we wrote that renal filtration would increase urinary creatinine. However, as you pointed out, the opposite relationship could occur if the bat has enough body water from feeding. In this case, renal filtration was to remove water to keep the water balance which would result in more urine production with low urinary creatinine concentration. This possibility could

be measured if we could have collected all urine from a bat from the beginning to the end of the one-day torpor. However, we could not make it. Although our spot collection of urine can reflect their daily average and the individual variation in urine production if they urinate only once when we captured them, this might not be the case. In addition, we could not make a precise measure of the volume of our urine sample (to see further reasoning refer to our response to the second request of reviewer #2 in the response letter as well). We integrated this point into the discussion, and we will keep this in mind for our future study.

Original sentences and revised sentences are as follows:

Original

“When bats decrease torpid time, body temperature and renal filtration rate will increase. Such an increase in renal filtration will result in the accumulation of creatinine in urine during daily roosting in the active season. Further study on the physiological ecology of bats in torpor will help us understand the direct causation of the change of urinary creatinine concentration during roosting in the active season.”

Revised

“When bats increase arousal time, renal filtration rate will increase as their body temperature increase. Such an increase in renal filtration in the daytime will result in water filtration or reabsorption in the nephron depending on individual body water balance. For example, bats with lower body water will increase water reabsorption which will then increase the accumulation of creatinine in urine. However, bats with enough body water will produce more urine by water filtration which increases urine volume and result in lower urinary creatinine concentration. As we did spot urine collection, this may not well represent entire urine production during day roosting. In addition, we could not precisely measure the volume of urine we collected, so these possibilities could not be tested in the current study. A controlled study that measures the volume of urine during daytime can clarify the direct causation of the change of urinary creatinine concentration during day roosting in the active season.”

Your comments (LL 288-289 and LL 290-291 and two deleted comments)

(LL 288-289) You do not account for any variable of surface area ratios. Realistically greater body mass might reduce surface area to volume ratios and potentially reduce EWL.

(LL 290-291) Body mass in the context you have measured it does not equate at all to metabolic rate. You have no information about this relationship and how this is effected by inert tissues such as fat- which is potentially what is causing the change in BM. The change in BM could also simply be an artefact of feeding as the increased BM is simply gut contents.

We left our responses to your deleted comments as well. See the attached document with track changes.

Our response and changes

(to LL 288-289) As you pointed out, the point that greater body mass can reduce surface area to volume ratios. This is the main thing we wanted to point out through this study. In the following sentences in the same paragraph, we discussed it. We also removed most of our discussion on the interaction term between the forearm length and body weight and revised our discussion (See LL 288-311). Thank you that you made us realize that the effect of body weight might not be that simple.

(to LL 290-291) You are right. We could not rule out other possibilities including those you pointed out. To make a more balanced discussion, we also integrated other possibilities (LL 288-311).

Original sentences and revised sentences are as follows (LL 288-311):

Original

“The significant interaction term in Model 2 makes it difficult to interpret our findings on the negative effect of body weight and the positive effect of forearm length on creatinine concentration during the active season. As seen in Fig. 5b, with an increase in body weight, urinary creatinine concentration decreases, particularly during daily roosting. Therefore, bats may lose body water from evaporation from the skin and/or urination which will then increase the body fluid concentration during daily roosting in the active season. To compensate for such water loss during roosting, bats need to increase renal water reabsorption. Bats with greater body weight can utilize stored energy more freely. As metabolism produces water that they can utilize to maintain body water balance¹³, heavier bats may not need more renal water reabsorption than lighter bats. On the other hand, the positive correlation between forearm length and creatinine concentration in Fig. 5d, may be due to an increase in the SVR of the bats with longer forearms but with lighter body weight. Across bat species, larger forearm length correlates to larger body size and weight^{17,48,49}. Therefore, the SVR will decrease with an increase in body size between bat species^{50,51}. However, within a species, the variation of SVR may be more complicated and reflecting individual feeding status or reproductive status. For example, pregnant females have higher wing loading (body mass / wing surface area) and this will decrease SVR⁵². As shown in Fig. 6a, body weight increased with an increase in forearm length from fed bats (caught at dawn) but it did not change from unfed bats (caught at dusk) during the active season. This difference between fed and unfed bats suggests that bats with longer forearms may lose more body weight during daily roosting in the active season. One possible reason for such difference may be that bats with longer forearms lose more water from their skin. This finding suggests that larger SVR can be a

determining factor of renal water reabsorption during daily roosting in the Eastern bent-winged bats. RH and temperature of the roosting site might be additional influencing factors. Unfortunately, we could not investigate this topic further because of a malfunction in our humidity logger (Fig. 2c). However, given that the roosting caves had running water throughout the study period, the humidity in them was probably high, and so cutaneous water loss might not have been an important factor influencing urinary creatinine concentration in the current study. The high urinary creatinine concentration (high urine concentration) from bats with lower body weight and longer forearms in the current study therefore suggests that individual variation in water stress exists depending on body weight and wing size.”

Revised

“As shown in Fig. 3b, urinary creatinine concentration decreases with an increase in the body weight in bats that emerged from the cave after day roosting. If bats with greater body weight can utilize stored energy for metabolism than lighter bats during day roosting, heavier bats may not need more renal water reabsorption than lighter bats due to more metabolic water production¹³. However, it is also possible that the heavier weight is a simple reflection of less defecation or urination caused by longer torpor or other reasons that resulted in lower urine filtration rates. The positive correlation between forearm length and creatinine concentration in Fig. 3d may be due to an increase in the SVR of the bats with longer forearms but with lighter body weights. Across bat species, larger forearm length correlates to larger body size and weight^{17,50,51}. Therefore, the SVR will decrease with an increase in body size between bat species^{52,53}. However, within a species, the variation of SVR may be more complicated and may reflect individual feeding status or reproductive status. For example, pregnant females have higher wing loading (body mass / wing surface area), and this will decrease SVR⁵⁴. It is also possible that bats with different forearm lengths but with the same body weight (or vice versa) might also face different water stress levels due to the difference in SVR. As shown in Fig. 4, we demonstrated that bats with heavier body weights produce more diluted urine (less creatinine concentration). This effect of body weight on urine concentration might also be influenced by forearm length as it can influence SVR. If this is the case, the high urinary creatinine concentration (high urine concentration) from bats with lower body weight and longer forearms in the current study suggests that individual variation in water stress exists depending on body weight and wing size. Although the effect of individual fluctuations in SVR on the body water balance within a species has been rarely studied so far, studying this topic will be important to understand the physiological adaptation of each bat species to water stress better.”

Your comments (L 314)

Can you really conclude its feeding? Exercise and urination probably also contribute. The change in fuel source would also play a role?

Our response and changes

Yes, you are right. We integrated your point in the next sentences as follows (LL314-316).

Variation in urinary creatinine depending on feeding status (capture time) from July to September (Fig. 1a) demonstrates that daily roosting may increase urine concentration, while night feeding (including drinking) activities decrease urine concentration (Fig. 1b). However other night activities such as urination and flight may be an influencing factor particularly in October when the prey items decrease.

Your comments (LL 316-318)

How come this difference is now considered to be small but before is considered to be large enough to warrant discussion about SVR?

Our response and changes

This was our mistake in writing in English. We wanted to point out that fed bats were heavier than unfed bats from July to September. In other words, the weight difference between fed and unfed bats in each month did not vary much over months from July to September. The original and revised ones are as follows (LL 316-319):

Original

“Given the small weight difference between fed and unfed from July to September seen in Fig S1a, the increase of urinary creatinine from July to September after daily roosting was not due to the low amount of feeding.”

Revised

“Body weight difference between fed and unfed bats in each month consistent from July to September – fed bats were heavier than unfed bats (Fig 1a, Supplementary Table 1). Therefore, the increase of urinary creatinine from July to September after day roosting was not due to the low amount of feeding.”

Your comments (L 364)

More details on how many bats were captured, what sex and how the samples were partitioned for analysis is needed!

Not just in the supplementary materials

Our response and changes

Thank you for your suggestions. We integrated those details in the result sections (LL 147-152) and move one table (previously Supplementary Table 1) to the main manuscript.

It's a pity that we could not measure all urine sample volumes. As volumes of urine samples were very small (usually 20 to 30 μ l), we could not precisely measure them with any of the volume measurement tools. We thought about some methods, but we found that they were not that accurate as well. For example, we thought of first measuring the weight of the urine-contained tube and measuring the weight of the empty tube to calculate the urine weight. However, as we don't know the density of urine, we could not precisely calculate the volume from weight. In addition, such a process needs thawing and transferring of samples that could influence the concentrations or stability of the substrates in the urine. Therefore, we decided not to do so.

Your comments (L 392)

Hibernating bats do not urinate. You must have waited some time for them to arouse from torpor before taking the sample. How long did you wait?

Our response and changes

When we conducted a study on the greater horseshoe bats, we waited up to 5 minutes (See Ryu et al 2021, *BMC Ecology and Evolution*). However, for bent-winged bats in the current study, when we detached them from a cluster, most of the bats just urinated. As you pointed out some bats did not urinate immediately and at the beginning of sample collection, we waited around 1 minute. We integrated this into the manuscript as follows. The reason for such a difference we thought was that bent-winged bats in the current studies hibernate in a large cluster and many of them started moving (getting a little aroused) when we start sampling collection. So many of them just urinated when we detached them as they had aroused already a bit. Most bats in the cluster did not fully arouse until we finished the sample collection. We integrated these points into the manuscript as follows (Red and underlined letters represent added parts).

“Urine sampling from hibernating bats was conducted immediately after detaching them from a cluster. Although most bats urinated right after we detached them, we waited about a minute for some bats to urinate while holding and petting their valley. If they did not urinate, we return them. We also did not conduct sample collection in February 2018 to reduce overall disturbance. All urine collection procedures in the hibernaculum took less than 1 hour.”

Your comments (LL 724-729 Fig. 2. previously Fig. 4)

As there is no difference between the sexes for creatinine concentrations why would you present the data as males and females when you show a difference between fed and unfed individuals. It

would be more interesting to partition the data in that way and pool the males and females over the hibernation season, and then move this graph to the supplements.

Our response and changes

Thank you for your suggestion. We thought about changing the figure (revised figure on the next page) following your suggestion and moving it to supplementary. However, the previous figure provides monthly trends, and we want to keep it. Providing such data is meaningful here as it directly lets us know that our statistical result that indicated no difference between sexes is reliable over months. We changed the figure title to include the creatinine difference between seasons as well as between sexes.

Response to Reviewer #2 (general comments)

Your comments

In this study, the authors measured and evaluated urinary metabolic wastes (creatinine and acetoacetate) in the Eastern bent-wing bat (*Miniopterus fuliginosus*) captured at a limestone cave and a mineshaft in South Korea. These sites were used by the bat population from spring to autumn and one of these sites also during the hibernation season. While there was no evidence of increased excretion of urinary ketone bodies during hibernation, seasonal variations in urinary creatinine indicated more concentrated urine production during hibernation. The interpretation and discussion of obtained results may provide some insights into the metabolic switch in utilization of glucose, proteins, and lipids and renal function in terms of glomerular filtration, reabsorption and urine formation preventing metabolite and water loss in insectivorous bats of the temperate zone in relation to their annual cycle of activity versus torpor and arousal bouts. Importantly, bats were sampled non-invasively and non-lethally.

The manuscript is rather well written in a readable way and can be recommended for publication following some improvements as suggested in following comments.

Our response

Thank you for your review and comments on our manuscript. As you pointed out, we also believe that our study on wild insectivorous bats may provide important insight to better understand the physiological adaptation for survival and reproduction. It is unfortunate that the methodological limitation made us not to be able to provide more concrete evidence of no waste of ketone bodies during hibernation in winter or the active season. Following your and reviewer #1's comments, we decided to mostly remove our discussion about the acetoacetate as we could not rule out the possibility of no detection of the acetoacetate due to our high dilution rate. We also revised some discussion about the interdependency between body weight and forearm length on urinary creatinine concentration. Previously, we thought that there was a significant effect of forearm length on the evaporative water loss, EWL. However, we mostly removed our suggestion and discussion on this matter. We did not measure the EWL directly and the relative humidity would be probably very high in the cave all year round. Therefore, we could not rule out the possibility that the statistically significant effect of forearm length on the urinary creatinine (so thus urine concentration) can be a simple correlation that is related to the size of bats rather than a real effect that indeed demonstrates a causative effect of the forearm length on the EWL. We hope that you will find out that the revised manuscript did not lose its importance to the field and meet your expectation for the revision.

Response to specific comment to Reviewer #2

1) Please provide argumentation for the selection of the bat species for the present study.

Our response

Thank you for your guidance. We integrated your point and basic information about the target species in the introduction to justify our selection of the target species (LL 119-129) as follows:

Eastern bent-winged bats are small insectivorous bats that are distributed in Korea, Japan, and China^{43,44}. They form a large cluster that consists of several hundred to thousands of individuals for hibernation. Pregnant females migrate for breeding and form a very large breeding colony⁴³. They are good study subjects to study physiological ecology in the wild. In particular, the subject population in this study forms a large colony of several hundred bats throughout the year, ensuring a large number of samples, which is often difficult in field studies. We can also capture them by hand for sampling without using any special equipment during winter as they form a large cluster at a hand-reaching distance from the ground. Although the amount of urine collected is small (around 20 to 30ul) in general, they also urinate easily during handling which helps us reduce the time required for urine sample collection.

2) Please provide data about collected urine volumes (only the number of urine samples mentioned in Table S1).

Our response

It's a pity that we could not provide the exact volume of urine samples as we could not measure urine volume precisely due to its small amount (usually 20 to 30ul). We could not find good volume measurement methods as well. We thought, for example, to measure the weight of a urine-contained tube first and then measure an empty tube to calculate the urine weight and then convert it into volume. However, as we don't know the density of each urine sample, we could not precisely calculate the volume from weight. In addition, as such a process needs thawing and transferring that could influence the concentrations or stability of the substrates in the urine, especially acetoacetate, we decided not to do it. During revision, we realized that knowing urine volume will provide important information to estimate the amount of urine production and its relationship to creatinine concentration. If we had that information, we could test whether the low creatinine concentration from heavier bats was due to an increase in urine production (volume), even though our spot collection of urine might not well represent the whole urine volume they produced during day roosting. We integrated this point into the discussion as well (LL 255 ~ 266).

3) On lines 370-371 you should be more specific about the urine collection procedure. In my opinion, the term genitalia is not correct. Wording of the sentence may also be understood as collection of urine after taking genitalia out of a cotton bag.

Our response

We revised the sentence on lines 384-385 from “We collected urine directly from their genitalia right after we took out them from the cotton bag.” to “We collected urine directly from bats right after we took out them from the cotton bag (Supplementary Fig. 3).” We added also a picture of our urine collection from a bat to supplementary Fig. 3.

4) Tools and their accuracy to measure body weight and forearm length are not mentioned.

Our response

We added the information in the method section as follows (LL 381-384).

We used an electronic scale (RE-260, CAS, South Korea) that is capable of measuring 2 to 500g with a minimum unit of 0.1g to measure body weight. For forearm length measurement, we used a digital calliper, CD-15APX (Mitutoyo Corp., Japan) with a minimum unit of 0.01mm.

Reviewers' comments:

Reviewer #1 (Remarks to the Author):

I am pleased to see that the authors have made some substantial changes to the body of the manuscript and have carefully considered the comments made by reviewers. I still feel as if some of my concerns were perhaps not completely understood by the authors and they still have some clarification to do regarding their methods and conclusions.

The heavy focus on the correlations with forearm length need to be much better explained, the sections in the discussion should be made more concise and the statistics used need to be included in the methods. Much of this is still speculative and I feel like the authors are trying to focus too hard on something that is potentially not meaningful from their results.

There was almost no focus on the extreme drop in creatinine concentrations in October and November prior to the increase during hibernation. It seems to me that including October in the active season and excluding November from the hibernation season is not justified, it simply leads to an easier ability to detect differences between the seasons. Therefore better justification of these delineations are needed.

I think the authors also need to acknowledge that their sample size in winter is very low and make it more clear to the reader in the discussion that the bats captured at dusk and dawn are presumed to be "fed" and "unfed" but you have no evidence to suggest that they are other than a difference in the average masses. I agree that this is very likely that they had fed prior to returning to the roost, but you cannot say that they actually DID feed, you can only infer.

I would like to see a revised version of the manuscript as I feel that a lot of good work has been done for this research it just could be presented much better as to receive the attention it deserves.

Our revisions and responses to the reviewers are as follows.

We have revised our manuscript in accordance with the suggestions and comment from the reviewer and editor. We also conducted a thorough English revision of the entire manuscript after integrating these comments and suggestions. For better readability, we made some minor corrections that were not pointed out by reviewers. Any corrections are highlighted (in yellow) in the manuscript, and a version of the manuscript containing all tracked changes is also attached. We hope that our revision will meet your expectations.

Reviewer #1 (general comments)

Your comments

I am pleased to see that the authors have made some substantial changes to the body of the manuscript and have carefully considered the comments made by reviewers. I still feel as if some of my concerns were perhaps not completely understood by the authors and they still have some clarification to do regarding their methods and conclusions.

The heavy focus on the correlations with forearm length need to be much better explained, the sections in the discussion should be made more concise and the statistics used need to be included in the methods. Much of this is still speculative and I feel like the authors are trying to focus too hard on something that is potentially not meaningful from their results.

Our response

Thank you very much for your invaluable contributions to our manuscript. This time, we have accepted most of your comments, and revised our manuscript based on your suggestions. We are very pleased to have you as a reviewer.

We have made an effort to reduce our discussion, particularly the parts that were indicated as speculative. For instance, we have largely removed our discussion about the possible effects of forearm length on the evaporative water loss. As you pointed out, our data may imply such a possibility, but it does not demonstrate any of the possibilities experimentally.

Additionally, we have included more information about the statistics we used and removed unnecessary information from the figures, particularly Fig. 3. We hope that the new figures are clear, concise, and better represent what we found in our two statistical models.

Your comments

There was almost no focus on the extreme drop in creatinine concentrations in October and November prior to the increase during hibernation. It seems to me that including October in the active season and excluding November from the hibernation season is not justified, it simply

leads to an easier ability to detect differences between the seasons. Therefore better justification of these delineations are needed.

Our response

We have included an explanation for not considering the November sample as part of the active season at LL 144-151. The details are as follows:

LL 144-151

In November, we could only capture five bats at dusk and collected urine from three bats, although we also conducted a mist net survey at dawn. The five bats were caught outside of the mist net at dusk implying that they were coming into the cave from outside, not going out from the cave. This was different from what we observed from July to October when most bats were caught inside of the mist net at dusk while they were exiting the cave. We did not observe any bats around our mist net at dawn in November as well. Therefore, we considered the active season to be from June to October, the hibernation season from December to March, and November a transient period.

Your comments

I think the authors also need to acknowledge that their sample size in winter is very low and make it more clear to the reader in the discussion that the bats captured at dusk and dawn are presumed to be "fed" and "unfed" but you have no evidence to suggest that they are other than a difference in the average masses. I agree that this is very likely that they had fed prior to returning to the roost, but you cannot say that they actually DID feed, you can only infer.

I would like to see a revised version of the manuscript as I feel that a lot of good work has been done for this research it just could be presented much better as to receive the attention it deserves.

All of my detailed comments are provided in the attached, marked up word document.

Our response

We have integrated your concerns regarding sample sizes and potential sampling bias towards males (LLs 242-246, 285-287, 292-297). Specifically, from line 292 to 297, we indicated the exact number of samples from males and females respectively and expressed our concern about such biased samplings. We also changed the terms "fed" and "unfed" to "captured at dusk" and "dawn" throughout the manuscript. Additionally, we revised figures and tables based on these changes. As you pointed out, certain aspects could be inferred but were not demonstrated by our data.

Lastly, we would like to express our appreciation for your comments and guidance. Your comments were invaluable, critical, and helpful in improving our paper. We have included our

responses to you in the following paragraphs, and you can also access them directly in the attached document with tracked changes. We hope that our revision will meet your expectations.

Response to specific comment to Reviewer #1

We have accepted most of the changes you made to the manuscripts and highlighted all modifications. Below, we provide our responses to your specific comments, which we have extracted from the manuscript. Please note that some of your comments lack line numbers as they were removed in the revised version. However, we have responded to all deleted comments in the attached manuscript with tracked changes. The line numbers referenced here correspond to the revised document, not the previously revised draft we submitted last June.

Your comment (L43 and one deleted comment)

1. I recommend checking out some recent work by Liam McGuire's group about the importance of EWL in hibernating bats. Perhaps a better reference than the older stuff. <https://pubmed.ncbi.nlm.nih.gov/34675252/>
2. Again see ref above.

Our response and changes

We have checked the papers from McGuire's group and incorporated their work into our references. We were previously unaware of their work on bats, and we appreciate your bringing it to our attention. Their research provides valuable insights for our work. We decided to remove the second sentence by integrating the point to the previous sentences, so your second comments no longer exist on the revised manuscript.

We have also revised our sentences for better flow. The original sentences, which included your correction, and the revised sentences are as follows (LL 41-43). Red and underlined letters represent the revised parts from the previous sentences.

Original: For bats, in particular, due to their small body size and greater surface-to-volume ratio (SVR) compared with other small mammals, hibernation can be challenging^{6,7}. Yet hibernation allows bats to reduce metabolism and water loss as an important adaptation to energy and water-saving and thus winter survival⁸.

Revised: Bats, in particular, exhibit various behavioral strategies and physiological adaptations to cope with challenges such as evaporative water loss during hibernation, due to their small body size and greater surface-to-volume ratio (SVR) compared to other small mammals⁶⁻⁹.

Your comments (LL 72-73)

This is, and much of this whole paragraph, suggests bats are inherently bad at maintaining water balance. The high death toll is not because they are bad at balancing their water it is because WNS infiltrates the wing membranes disrupting their ability to balance cutaneous WL.

Our response and changes

We agree with your point. Even if bats exhibit higher SVR, it does not necessarily imply that they struggle with maintaining water balance. We realized we had a hidden assumption that bats are poor at balancing their body water. However, this is not accurate. We revised this paragraph by removing parts that could potentially suggest that bats struggle with water balancing.

We revised sentences (LL 43-45) from:

“However, maintaining water balance during hibernation is still challenging, as seen in the massive bat population losses in North America due to white-nose syndrome²¹⁻²³.”

to

“However, relying on such abiotic factors may not be sufficient for maintaining water balance during hibernation, necessitating behavioral and/or physiological solutions^{6,8,9}.”

Your comments (L75 deleted)

You’ve already mentioned this.

Our response and changes

Yes, we did. We accepted your correction (deletion of drinking).

Your comment (LL 160-161)

Please tell us what the variations were provide the min and max temperatures in the text.

Our response and changes

We updated Mean \pm SD and the total temperature ranges following your suggestions. Please check our updates at LL 160-167.

Your comment (LL 165-166)

Provide details of the RH and its variations in the text. Summary statistics at the very least.

Our response and changes

We updated those values on the manuscript as follows.

Original:

The average daily RH also varied between the locations of the cave.

Revised:

The average daily RH also varied between the locations within the cave (entrance: 72.28 ± 17.24 %, range of 33.4–97.9%, middle: 64.86 ± 14.9 %, range of 38.0–87.2%, end: 76.77 ± 24.16 %, range of 35.5–98.5%).

Your comment (LL 174-184)

Again can we have done actual values here. Some summary statistics of the mean masses or at least the differences between seasons. Anything to support your words that doesn't require us to only decipher your results from the figures.

Our response and changes

Thank you for your suggestions. We have now included that information. One detail we could not include is the body mass of the bats captured in the hibernation season. We did not measure their body mass or forearm length to minimize our time in the hibernaculum.

We updated those values on the manuscript as follows.

Bats captured at dawn (51 in total) upon their return to the roost were heavier than those captured at dusk (68 in total) during the active season, with masses of 16.03 ± 1.42 g and 14.81 ± 1.34 g respectively. A Welch's *t*-test ($t = 4.78$, $df = 104.56$, $p < 0.001$) demonstrated this difference is statistically significant. However, bats captured at dawn were not always heavier than those captured at dusk, as observed in October (refer to Supplementary Table 1; Fig. 1a). While there was a decreasing trend in the body mass of bats captured at dusk until September, this trend reversed in October. The forearm length, which serves as a proxy for surface area, showed no significant difference between bats captured at different times (Welch's *t*-test; $t = -0.29$, $df = 99.14$, $p = 0.77$), indicating that our data did not exhibit a significant sampling bias for wing size between different capture time (Supplementary Table 1).

Your comment (L 176)

What is this? The *t* value and *df*?

Our response and changes

Oh, we missed it. Yes, it is.

Your comment (L 189 deleted due to revision)

Unless you take into consideration a gravitational constant you are measuring mass not weight.

Our response and changes

Following your suggestion, we changed all body weight to body mass.

Your comment (L 190)

Better to just refer to the capture times than your assumptions of feeding status until the discussion.

Our response and changes

You are right. We also deleted fed or unfed from all figures as well.

Your comment (L 195)

Really 0.065?? That's very very low.

Our response and changes

Yes, it is very low. In the model, all predictor variables are categorical, and this is a full model fitting result in which two variables (sexes, locations) were non-significant – only a difference of season was found. In addition, it was 0.065 when the samples collected in November were included in the analysis. It was an error we made due to our data coding. Based on our definition of season (LL 143-151), these data points from November should be removed. After removing them, R^2 becomes 0.101. Although it is still low, given the significant individual variations in the urinary creatinine concentration, it makes sense. Nevertheless, we are mindful of such weak R^2 values and have toned down all of our discussion of our findings.

Your comment (L 196)

Where did this value come from? I cannot find this in the table.

Our response and changes

The results were obtained from the “summary” function in R, which assesses the significance of the model – how well the full model with 3 independent variables fits with the data set. We followed a commonly used model presentation method in primatology where the first author completed his graduate course.

<https://besjournals.onlinelibrary.wiley.com/doi/full/10.1111/2041-210X.12577>

If there's a better way to present this, please let us know.

Your comment (LL 207-209)

I don't think this is what your data show. Only October seems to have any true positive linear correlation but without the stats to back this up the statement and the figures are misleading.

Our response and changes

We have taken your concerns into account and revised our sentence accordingly. We have also removed all graphical indications of monthly variations in creatinine concentration in Fig 3. Additionally, we revised the figure order and changed the numbering following our revision. As a result of these changes, we significantly revised LL 200-211. We also removed interpretations of our results in the result section, as many of them should be in the discussion section. You can also see your deleted comments and our response to them in the attached document with tracked changes.

Your comment (LL 213-214)

Or this could be the other way around....

Our response and changes

You are right. We revised this part by integrating your point.

Original:

Despite the aforementioned significant main effects in Model 2, the significant interaction term suggests that the effect of body mass on urinary creatinine might vary depending on one's forearm length.

Revised:

In addition, despite the aforementioned significant main effects in Model 2, the significant interaction term suggests interdependency between the effect of body mass and forearm length on urinary creatinine.

Your comment (LL 216-219)

Where is this in the results for model 2?

Our response and changes

This was a visual investigation of the Model 2 using an R package called “interactions”. To avoid the risk of false-positive findings, we visually investigated the interaction terms instead of conducting post-hoc tests. Our revision is as follows.

Original:

To clarify this dependence between the two variables we investigated how the effect of body weight changed depending on forearm length during daily roosting and night activities in Model 2 by fitting the effect of body weight with three forearm lengths (mean forearm length and mean \pm 1SD).

Revised:

To clarify this interdependence and its potential effect on the urinary creatinine, we conducted a post-hock investigation of the effect of body mass on urinary creatinine concentration relative to forearm length in Model 2. We visualized the interaction term of Model 2 by fitting the effect of the body mass on urinary creatinine with three different forearm lengths (mean forearm length and mean \pm 1SD) – simple slopes analysis⁴⁴.

Your comment (L 222)

I understand that you are assuming an interaction between forearm length and body mass- this is a justified assumption. The idea that forearm length correlates with surface area needs to be supported by some additional literature. But I really do not understand what your method here is doing. It seems like you’ve assigned forearm lengths to each bat... not the forearm length that they had- so this is not a result of empirical data it is the result of a simulation??

I have now checked your methods section and this analysis is not listed anywhere and the details regarding the linear models presented in this figure and Fig 3 need to be explained.

Our response and changes

Thank you for your comment on this matter. We realized that we omitted the explanation of this analysis. It was a post-hock analysis of Model 2, conducted using the “interactions” package in R. The forearm data and body mass are perfectly matched to each individual.

It is important to note that we did not make any inferences using the estimates and p values from the post-hoc analysis (resulting from simple slope analysis) as it could increase the chance of false-positive findings. Instead, we provided figures showing how the effect of body mass on urinary creatinine could vary depending on forearm lengths and bat-captured time. However, we realized that we did not include three-way interactions between body mass, forearm lengths and captured time, so we revised Fig. 4 to show only the effect of the interaction term between body mass and forearm length on urinary creatinine.

Lastly (although we mentioned it earlier to respond to your comment on L 196), Fig 3 was initially intended to provide more visual information for Model 2. We followed a commonly used method to present a mixed model (Zuur & Ieno, 2016, *Methods in Ecology and Evolution*, DOI: 10.1111/2041-210X.12577). However, as you pointed out, the regression lines in Fig 3, did not perfectly match with the results from Model 2 as the figure used a simple regression but Model 2 used a GLMM. Therefore, we decided to remove most of that visual information from Fig 3 to make it easier for readers to understand the figure and Model 2. Now, Fig. 3 consists of only two regression lines that were calculated with two different simple regressions, and these are indicated in the figure legend.

Your comment (L 226)

At least give the number here.

Our response and changes

We previously inserted CV numbers in the methods section. Now we have moved it here.

Your comment (LL 234-246)

You need to acknowledge your very small hibernation sample set and the likely bias due to the grave differences in sample sizes.

Our response and changes

You are right. We integrated your points at the end of the paragraph as follows (LL 242-246).

Nevertheless, there was a large difference in the number of urine samples between the hibernation season and active season in this study. This discrepancy in sample size between the two seasons could introduce a sampling bias and should be taken into account before drawing comprehensive conclusions on this topic.

Your comment (LL 285-287)

Or may simply arise from our very small sample size.

Our response and changes

Yes, you are right. We integrated your points at the end of the sentence as follows (LL 286-287).

This difference may reflect a difference in reproductive physiology between the two species, although we could rule out sampling bias due to the small number of samples in the current study.

Your comment (LL 287-292)

What does this have to do with urine production? Males are arousing more frequently than females?

Our response and changes

Yes indeed, in the case of greater horseshoe bats, males may arouse more frequently than females, which could lead to sex differences in urine production and creatinine accumulation in urine. We have incorporated this point at the end of the sentence (LL 287-292) as follows.

Ovulation occurs after hibernation in greater horseshoe bats (delayed ovulation), and mating probably continues during hibernation^{46,47}. In contrast, ovulation occurs before hibernation in Eastern bent-winged bats (delayed implantation), and thus mating during hibernation is unlikely⁴⁸⁻⁵⁰. Therefore, males in the two species may exhibit different arousal frequencies and this can be a possible explanation for such a species difference in urinary creatinine during hibernation.

Your comment (LL 292-298)

This whole part could be condensed to only 3 sentences.

Our response and changes

We reduced this part as follows.

Original:

There is not enough data to determine whether there is a sex difference in creatinine concentration during the active season, as the number of urine samples from females was low in July and August. Eastern bent-winged bats form maternity colonies where females aggregate and rear their pups together⁴³. As female bats in the current study form a maternity colony at another location, urine samples collected in July and August were male-biased. We cannot rule out, therefore, the possibility that inter-sex difference exists in the urine concentration resulting from the difference in body water utilization during the active season depending on their reproductive status. Lactating females may need more water and nutrients for milk production⁴⁹. We lack data for lactating females in July and August. It will be important to investigate in future studies whether lactating females face more water and energy shortage that increase urinary concentration and ketone bodies. This will help us to better understand sex effects on water stress depending on reproductive status.

Revised:

Although we could not find a sex difference in creatinine concentration during the active season, there was significant sampling bias, particularly in July (1 female urine sample vs 25 male urine samples) and August (5 vs 29). Such sampling bias makes it difficult to rule out a possible sex difference in urinary creatinine during active season which could be influenced by female reproductive events including nursing⁵¹. Further investigation is necessary to clarify whether such reproductive events can result in a difference in water stress between sexes.

Your comment (LL 294-295)

You only have 2 samples from females in December... how is that better?

Our response and changes

You are correct.

However, in December, we only had five male samples, and the difference is not as pronounced as it is in July and August. Our discussion primarily focuses on the results from Model 2. However, following your suggestion, we have expanded our discussion on potential sampling bias.

Your comment (LL 299-301)

Fig 3d either shows basically a flat line or a negative correlation... only in October do the data suggest a positive linear correlation. If you are going to stick with this argument of SVR you really need to rethink how you present the data because separating the data by month needs to be clarified in the methods and your model selection here needs to be explained. These are not the results from model 2 if I understand your Table correctly. You also do not clarify in your methods that month was considered as a factor in any of the analyses- just season...

Our response and changes

We have taken your points into consideration and removed most of our discussion that was based on our visual investigation of Model 2. We found it to be confusing. The previous Fig 3b and 3d provided a possible interaction effect of capturing time and body mass on the urinary creatinine and the effect of forearm length on the urinary creatinine with monthly trends respectively. However, as you pointed out, such a monthly effect should also be tested in the model for a more accurate conclusion. As we mentioned earlier (our response to your comment on L 222), we did not conduct simple regression tests. They were drawn by the simple regression line calculation function in R, more precisely within the ggplot2 packages. It was to follow a commonly used method in primatology. However, they are not relevant as we have removed those details from the figure and removed our discussion on them. We hope that our revision is now concise and addresses your concerns.

Your comment (L 302)

This is also mass- these papers refer to mass.

Our response and changes

We decided to use mass in the entire manuscript.

Your comment (LL 303-312)

From here to the end of the paragraph needs to be written more concisely. It is long winded and also of speculation without clear evidence from your results to support it. I understand you want to talk about surface area to volume ratios but again- you need to do a better job of investigating this effect in your results rather than just assuming that the fact you found a significant interaction shows this. Because a lot of your linear regressions are very low R² and very flat.

Our response and changes

We revised this part following your sentence. As you pointed out, our study could not provide concrete evidence of the effect of forearm length on the urinary creatinine that resulted from increased evaporative water loss. Therefore, we removed most of previous discussion based on the weak correlation. Thank you very much for your critical comments that help us avoid overinterpretation that could be incorrect.

Original:

However, within a species, the variation of SVR may be more complicated and may reflect individual feeding status or reproductive status. For example, pregnant females have higher wing loading (body mass / wing surface area), and this will decrease SVR⁵⁴. It is also possible that bats with different forearm lengths but with the same body weight (or vice versa) might also face different water stress levels due to the difference in SVR. As shown in Fig. 4, we demonstrated that bats with heavier body weights produce more diluted urine (less creatinine concentration). This effect of body weight on urine concentration might also be influenced by forearm length as it can influence SVR. If this is the case, the high urinary creatinine concentration (high urine concentration) from bats with lower body weight and longer forearms in the current study suggests that individual variation in water stress exists depending on body weight and wing size. Although the effect of individual fluctuations in SVR on the body water balance within a species has been rarely studied so far, studying this topic will be important to understand the physiological adaptation of each bat species to water stress better.

Revised:

However, inter-individual variations in SVR within a species may be more complex as an individual's body mass changes over time depending on several factors such as pregnancy or feeding⁵⁶. As shown in Fig. 4, Eastern bent-winged bats with heavier body masses produce more diluted urine. This negative effect of body mass on urine concentration changes with the forearm length. We were unable to test whether this interdependence between forearm length and body mass was due to an increase in SVR in bats with longer forearms. A well-controlled experimental study will help clarify whether water stress varies depending on an individual's body mass and wing size. Further investigation into this topic will also enhance our understanding of the physiological adaptations of each bat species to water stress.

Your comment (L 335)

Tb? What is TB? Do you mean RH?

Our response and changes

Oh, our mistake. It should be RH.

Your comment (L 375)

What is physiological measurement?

Our response and changes

It was body mass and forearm length measurement. We revised it as follows.

When they did not urinate, we carefully patted their abdomen several times. If these attempts failed, we refrained from urine sampling and started measurement of body mass and forearm length and banding.

Your comment (L 416)

Why did you log transform the results?

Our response and changes

It was to make the data follow a normal distribution. We added an explanation at LL 417-419 as follows:

A natural log-transformation was applied to the urinary creatinine concentration to normalize the data distribution.

Your comment (L 704 Fig. 1)

As I mentioned in the text I really think you should refrain from calling these fed vs unfed. Dusk and Dawn are the correct details you know and any inference about whether or not the animals had fed or not is speculative. It does appear that animals have fed in the dawn samples but you cannot say that for sure so all reference should be to the sample state and you can discuss the potential feeding in your discussion.

Our response and changes

Your point is very valid, and we changed all instances of “fed” and “unfed” to “dusk” and “dawn” following your comments. We also adjusted the y-axis of Fig 1b to make it identical to the y-axis of Fig 2 following your comments. Please check the revised figure below.

Your comment (L 712 Fig. 2)

Be consistent with the formatting of your graphs- the x axis should be the same between figure 1 & 2 and the y axis for creatinine concentrations should have the same limits.

Our response and changes

We revised both figures to have the same range. Check the revised figure below and above on this page.

Your comment (L 728 Fig. 3)

Where are the methods for all the statistics performed to get you to these graphs!?

Our response and changes

As we explained earlier (our response to your comments L 222 and LL 299-301), they represent the result of Model 2. We revised the figures to represent Model 2 more precisely as follows and added an explanation of the two regression lines were derived from simple regression. Please check the previous and the revised figures below.

Previous Fig 3

Revised Fig 3

REVIEWERS' COMMENTS:

Reviewer #1 (Remarks to the Author):

Thanks you for the hard work at revising your manuscript! I find it very much improved. I only have noted 2 small errors in the text that require fixing.

Abstract Line 17/18 - change order of the sentence "are crucial for small mammals, including bats, to maintain energy and water balance."

Intro Line 39 - limited fat storage capacity.

Our revisions and responses to the reviewers are as follows.

We are deeply appreciative of the reviewers and you for the invaluable assistance and input with our manuscript from the initial draft to its current form. We have integrated the two suggestions from Reviewer #1. We also revised and integrated the points following the final revision instruction. Any corrections are highlighted (in yellow) in the manuscript, and a version of the manuscript containing all tracked changes is also attached. We hope that our revision will meet your expectations.

Reviewer #1 (Remarks to the Author):

Thanks you for the hard work at revising your manuscript! I find it very much improved. I only have noted 2 small errors in the text that require fixing.

Our response

Thank you for your continued support with the manuscript. We really appreciate your help with the manuscript and invaluable contributions to our manuscript. We have integrated your two suggestions in to our manuscript.

Abstract Line 17/18 - change order of the sentence "are crucial for small mammals, including bats, to maintain energy and water balance."

> We revised it as you suggested.

Intro Line 39 - limited fat storage capacity.

> We added "fat" there as you suggested.